



# Measuring Snow Specific Surface Area with 1.30 and 1.55 µm Bidirectional Reflectance Factors

Adam Schneider[1], Mark Flanner[1], and Roger De Roo[1]

[1]Department of Climate and Space Sciences and Engineering, Climate & Space Research Building, University of Michigan, 2455 Hayward St., Ann Arbor, MI 48109-2143

**Correspondence:** Adam Schneider (amschne@umich.edu)

**Abstract.** Snow specific surface area (SSA) is an important physical property that directly affects solar absorption of snow cover. Instrumentation to measure snow SSA is commercially available for purchase, but these instruments are costly and/or remove and destroy snow samples during data collection. To obtain rapid, repeatable, and in situ surface snow SSA measurements, we mounted infrared light emitting diodes and photodiode detectors into a 17 cm diameter black styrene dome. By flashing light emitting diodes and measuring photodiode currents, we obtain accurate 1.30 and 1.55 micron bidirectional reflectance factors (BRFs). We compare measured snow BRFs with X-ray micro computed tomography scans and Monte Carlo photon modeling to relate BRFs to snow SSA. These comparisons show an exponential relationship between snow 1.30 micron BRFs and SSA from which we calculate calibration functions to approximate snow SSA. The techniques developed here enable rapid retrieval of snow SSA by a new instrument called the Near-Infrared Emitting and Reflectance-Monitoring Dome (NERD).

## 1 Introduction

Earth's surface albedo is a primary component of the planetary energy budget. Of the vast natural surface types that determine Earth's fundamental radiative properties, snow cover is the most reflective. Fresh snow cover is especially reflective in the visible and less so in the near-infrared spectra, reflecting as much as 90 percent of the direct solar irradiance into the upward facing hemisphere. Snow cover is also a highly dynamic, unstable surface type in the Earth system. Changes in snow albedo, for example, drive positive albedo feedback and other nonlinear processes that can enhance snow melt and surface temperature anomalies (Fletcher et al., 2012; Qu and Hall, 2007; Winton, 2006; Hall, 2004). Positive snow internal albedo feedback occurs due to the strong dependence of snow infrared reflectance on snow specific surface area (SSA). The Snow, Ice, and Aerosol Radiation (SNICAR) model (Flanner et al., 2007) demonstrates this dependence and is applied here to simulate the spectral black-sky albedo of nadir illuminated snow in Fig. 1.

Snow SSA is defined as the total ice surface area to mass ratio, such that

$$\text{SSA} = S/M = \frac{S}{\rho_{ice}V},\tag{1}$$

where $S$ is the total surface area of a mass $M$ of snow occupying an ice volume $V$ and $\rho_{ice}$ is the density of ice (917 kg m$^{-3}$) (Legagneux et al., 2002; Gallet et al., 2009). Previous studies demonstrate the strong dependence of snow infrared reflectance



on snow SSA (Domine et al., 2006; Gallet et al., 2009). Modeling studies, such as those from Wiscombe and Warren (1980) and Flanner et al. (2007), also demonstrate this strong dependence using sphere effective radius as an optical metric for snow grain size. Gallet et al. (2009) also quantify snow SSA by its sphere effective radius ($r_{eff}$), defined by the radius of a sphere having the same surface area to volume ratio as the particles, such that

$$\text{SSA} = \frac{3}{\rho_{ice} r_{eff}}. \tag{2}$$

Other studies quantify snow grain size by its sphere effective radius (RE) as it relates to the projected area of a particle, so that

$$\text{RE} = \frac{3}{4}(V/A), \tag{3}$$

where $A$ is the particle projected area (Jin et al., 2008). These expressions of sphere effective radii, $r_{eff}$ and RE, defined by ice particle surface area $S$ versus ice particle projected area $A$, respectively, are equivalent for convex bodies (see Appendix A).

As surface temperatures increase, snow albedo generally decreases as snow SSA decreases. Recent studies verify this process of natural snow metamorphosis on seasonal timescales in Antarctica (Libois et al., 2015), New Hampshire (Adolph et al., 2017), and Colorado (Skiles and Painter, 2017). Libois et al. (2015) find that SSA evolution occurs slowly in the extremely cold Antarctic environment. Adolph et al. (2017) monitor the evolution of snow albedo across three winter seasons in New Hampshire to determine a strong dependence of snow broad-band albedo on optically derived snow grain size ($r_{eff}$). These

observational studies inform us on snow albedo measurements conducted on clean snow, with small concentrations of light absorbing impurities (LAI) such as dust and black carbon (BC). Skiles and Painter (2017) observe seasonal scale snow albedo decline in springtime Colorado. In contrast, however, they find that snow albedo is primarily related to dust concentration. LAI can directly reduce snow albedo, but also indirectly darkens snow during metamorphosis. This indirect effect is demonstrated by Hadley and Kirchstetter (2012), where the albedo reduction due to the presence of BC in snow is amplified in snow of lower

SSA. This enhancement of snow albedo reduction is another source of instability in the snow pack that increases the strength of snow internal albedo feedback.

    When snow SSA decreases, a positive albedo feedback can exist where solar heating induces grain growth, further decreases SSA, and causes the snow surface to absorb additional solar radiation. Surface warming can also reduce snow grain growth rates, however, if growth processes from vapor diffusion and strong temperature gradients are affected negatively (Flanner

and Zender, 2006). Recent studies use X-ray computed microtomography (X-CT) to monitor the evolution of snow SSA in a high-temperature gradient (Wang and Baker, 2014) and in isothermal snow metamorphosis (Ebner et al., 2015). Ebner et al. (2015) show that measurements of snow SSA evolution in isothermal snow agree with the isothermal snow metamorphosis modeling framework developed by Legagneux et al. (2004) and Legagneux and Domine (2005). These studies express snow SSA in isothermal metamorphosis as function of time $t$ as follows,

$$\text{SSA} = \text{SSA}_0 \left( \frac{\tau}{\tau + t} \right)^{1/n}, \tag{4}$$

for initial snow $\text{SSA}_0$ at $t = 0$ and adjustable parameters $\tau$ and $n$.





Previous studies establish techniques to accurately obtain snow SSA using methane gas absorption (Legagneux et al., 2002), contact spectroscopy (Painter et al., 2007), infrared hemispherical reflectance (Gallet et al., 2009; Picard et al., 2009; Gallet et al., 2014; Gergely et al., 2014), and X-CT in cold rooms (Pinzer and Schneebeli, 2009; Wang and Baker, 2014; Ebner et al., 2015), but these methods require expensive, heavy equipment and measurements can be time consuming. Further, previous methods require that snow samples are collected and possibly even destroyed during measurements, preventing in situ snow observations over the span of just several hours. Because of the strong dependence of snow albedo on snow SSA (Adolph et al., 2017), the ability to obtain rapid, repeatable measurements that can describe the snow surface in basic physical terms is widely sought after.

Here, we introduce a new technique to measure snow SSA in a nondestructive manner using 1.30 and 1.55 µm bidirectional reflectance. By gently placing the Near-Infrared Emitting and Reflectance-monitoring Dome (NERD) onto the snow surface, multiple 1.30 and 1.55 µm bidirectional reflectance factors (BRFs) are obtained in just minutes with minimal alteration of the snow surface. To calibrate with respect to snow SSA, we compare snow BRFs with X-CT derived SSA to identify the exponential relationship between SSA and snow 1.30 µm BRFs.

## 2   Instrumentation and Methods

### 2.1   The Near-Infrared Emitting and Reflectance-Monitoring Dome

The NERD is designed to measure 1.30 and 1.55 µm BRFs. Four light emitting diodes (LEDs) and four photodiodes are mounted into a 17 cm diameter black styrene half-sphere (see Fig. 2). Two LEDs with peak emission wavelengths of 1.30 µm are mounted at nadir and ten degrees relative to zenith while two LEDs with peak emission wavelengths of 1.55 µm are mounted at 15 degrees off nadir. 1.30µm LEDs have spectral line half widths of 85 nm and half intensity beam angles of ten degrees, while 1.55 µm LEDs have half-maximum bandwidths of 130 nm and 20 degree beam angles. These high powered infrared LEDs are selected to illuminate a small oval of the experimental surface to maximize the reflected radiance signal. The reflected radiance signal is measured using four InGaAs photodiodes mounted in two different azimuthal planes; two each at 30 and 60 degrees relative to zenith. Photodiodes highly sensitive to light ranging from 800 to 1750 nm and relatively large active areas (1 mm) are selected to maximize sensitivity.

The NERD is similar to that of the DUal Frequency Integrating Sphere for Snow SSA measurements (DUFISSS) (Gallet et al., 2009) in that it uses 1.30 (1.31 in DUFISSS) and 1.55 µm emitters to illuminate the snow surface from nadir (15 degrees off nadir for 1.55 µm in NERD). LEDs are toggled using a Ruggeduino-ET (Extended temperature, operational down to -40 degrees C.; www.rugged-circuits.com/microcontroller-boards/ruggeduino-et-extended-temperature-40c-85c) connected to a LED driver. The LED driver generates an 80 mA square wave through each LED individually with a pulse width of two seconds (20 % duty cycle). The main distinction between the DUFISSS and the NERD is the type of reflectance measured. Gallet et al. (2009) use an integrating sphere to measure hemispherical reflectance. Here, rather, we direct photodiodes toward the illuminated surface in a black dome to measure BRFs. The interior of the dome is painted with a flat black paint to increase absorptivity and minimize internal reflections between the dome and snow surface. To detect reflected radiance signals,



photodiodes are reverse biased to induce currents linearly related to the amount of light incident on its active region. Because these light signals are reflected from the experimental surface, the currents induced by the photodiodes are very small (nano- to micro-Amps). To measure the small currents, the photodiodes are connected to transimpedance amplifiers (as in Fig. 2). The transimpedance amplifier circuits convert and amplify the small photodiode currents into measurable voltage signals. Finally,

an active low pass filter is installed between the amplifier and the analog-to-digital converter (ADC) to reduce noise. This filter is designed to have a time constant of less than 0.5 seconds to achieve balance between adequate noise reduction and speed. Waiting 0.75 seconds after toggling the LED allows for enough time for the photodiode current to stabilize. After these currents stabilize, 100 voltage samples (ranging from 0.1 to 1.0 Volts) are then rapidly collected using the Ruggeduino-ET's ADCs. The average voltage obtained during active illumination is differenced from the average dark current voltage to derive reflectance

factors. Because the orientation of LEDs and photodiodes are fixed, reflectance factors can be obtained after calibration using two diffuse reflectance targets in a manner similar to that used by Gallet et al. (2009), Gergely et al. (2014), and Dumont et al. (2010). These Lambertian targets reflect incident light according to Lambert's cosine law and appear equally bright at all viewing angles. The reflectance of the targets are measured with high precision across a broad spectrum. At 1.30 (1.55) µm, the white and gray targets have calibrated reflectances of 0.95073 (0.94426) and 0.42170 (0.41343), respectively, as reported

by the manufacturer. By comparing the measured voltage signal from the experimental (snow) surface to that measured from the reflectance targets, two BRFs at both 30 and 60 degree viewing angles are obtained for each light source. This procedure enables simultaneous measurements of multiple BRFs at 1.30 and 1.55 µm.

     To validate NERD reflectance measurements, we assess its measurement accuracy, precision, and responsivity by measuring BRFs of reflectance standards after calibration. Using both reflectance standards, we record ten BRF ($R$) measurements for

each LED / photodiode viewing zenith angle ($\theta_i; \theta_r$) combination during outdoor temperatures between -20 ° and +2 °C. In general, NERD BRFs of the Lambertian reflectance standards are accurate to within +/- 2 %. We quantify instrument precision (2 %) by computing root mean squared (RMS) errors from repeated measurements (see Table 1). Linear regressions quantify the linear responsivity ($A$) over the reflectance range of 0.41 to 0.95. Responsivity error ranges from -2 % to +3 % and from +1 % to +3 % at 1.30 and 1.55 µm, respectively. These results validate the NERD's ability to obtain precise BRFs with a

measurement uncertainty of 1-2 %.

## 2.2  X-ray Microcomputed Tomography

Snow BRFs measured by the NERD are complemented by X-CT scans. X-CT scans of snow are conducted at the U.S. Army's Cold Regions Research Engineering Laboratory (CRREL) in Hanover, New Hampshire. The machine is housed in a cold lab kept below 0°C allowing for X-CT of snow without significant melt.

Small samples of snow are collected in roughly 10 cm tall cylindrical plastic sample holders and placed into the machine. An X-ray source is emitted at 40-45 kV and 177-200 micro-Amps. X-ray transmittance is measured as the machine rotates the sample. Setting the exposure time to 340 ms at a pixel resolution of 14.9 µm with rotation steps at 0.3-0.4 degrees allow for fast scan times of roughly 15 minutes. These short scan times are necessary to complete the scan without too much absorbed radiation melting the snow. Processing software allows for samples to be reconstructed while computing physical properties





of which SSA are derived (Pinzer and Schneebeli, 2009). Three dimensional visualization software is used to generate images shown in Fig. 3.

## 2.3 Snow Samples

Snow samples for NERD snow SSA calibration were collected over the span of three years (winters 2015-2017). Measurements
of these samples were conducted during the months of February and March in 2016 and 2017 in Hanover, New Hampshire.

### 2.3.1 Fresh samples from 2016

Fresh snow samples were collected during a late winter snow fall event in 2016 just outside of Hanover, New Hampshire. Fresh snow from two different locations were scooped into coolers and then transported to the CRREL for analysis. Visual inspection of these samples revealed snow that appeared softer and less dense than the class of old samples. Because the surface
temperature was close to $0°$ C., the samples appeared to be wet. X-CT scans (Fig. 3a.) confirmed snow that was of relatively medium density ($350 \, \mathrm{kg \, m^{-3}}$), medium porosity (62 %), and medium SSA ($19 \, \mathrm{m^2 kg^{-1}}$).

### 2.3.2 Artificial ice crystals grown in a cold lab

One of the snow samples included in the NERD snow SSA calibration was grown inside a cold lab at $-20°$ C. using a forced temperature gradient. Analysis on this sample was conducted during winter of 2016. Visual inspection revealed a hardened ice
medium with a well defined crystalline structure. X-CT scans (Fig. 3b.) showed jagged ice micro-features of relatively medium density ($320 \, \mathrm{kg \, m^{-3}}$), medium porosity (65 %), and low SSA ($9 \, \mathrm{m^2 kg^{-1}}$).

### 2.3.3 Old sintered samples from 2015

The oldest class of snow samples used for the NERD calibration were collected during the 2015 winter season in Hanover, New Hampshire. These samples were then stored in a cold laboratory for a year at the CRREL at approximately $-20°$ C. During
February of 2016, visual inspection revealed snow that was highly sintered. As expected, X-CT scans (Fig. 3c.) confirmed that these two samples, distinguishable only by the container they were stored in, were of relatively high density ($610 \, \mathrm{kg \, m^{-3}}$), low porosity (33 %), and low SSA ($9 \, \mathrm{m^2 kg^{-1}}$).

### 2.3.4 Fresh needles collected during the March 14 snow storm

On March 14, 2017, a heavy daytime snow fall event in Hanover, New Hampshire enabled rapid collection and analysis of snow
samples. Cylindrical X-CT sample containers were placed in snow already on the ground. Snow fall filled sample containers in just a couple hours. Sample containers were carefully moved (with gloves) into coolers. Coolers were then rushed directly into the nearby lab for X-CT analysis. X-CT scans (Fig. 3d.) confirmed needle like ice structures. These structures presented a snow pack of relatively low density ($110 \, \mathrm{kg \, m^{-3}}$), high porosity (88 %), and high SSA ($66 \, \mathrm{m^2 kg^{-1}}$).



### 2.3.5 Fresh samples collected shortly after February (10-16) 2017 snow fall events

Moderately fresh snow samples were collected in the first couple days following snow storms in February 2017 in Hanover, New Hampshire. A few of these samples include snow with small amounts of added dust and BC. All samples with added LAI included in the NERD SSA calibration dataset were first screened to remove samples with heavy LAI loads that caused

direct snow darkening at 1.30 and 1.55 μm. Snow samples were shoveled into coolers and transported to the CRREL for X-CT analysis. X-CT scans (Fig. 3e.) revealed snow of relatively low density (170 kg m$^{-3}$), high porosity (82 %) and medium-high SSA (54 m$^2$kg$^{-1}$).

### 2.3.6 Samples collected after apparent metamorphosis on February 17 2017

After visibly apparent snow metamorphosis, partially aged snow from Hanover, New Hampshire was collected and transported

to the CRREL for X-CT analysis. Some of these samples include snow with added LAI. Samples with added LAI had shown visible signs of dramatic metamorphosis. X-CT scans (Fig. 3f.) confirmed these observations, revealing snow of relatively medium density (310 kg m$^{-3}$), medium porosity (66 %), and medium SSA (23 m$^2$kg$^{-1}$).

### 2.4 Monte Carlo Modeling of Bidirectional Reflectance Factors

The Monte Carlo method for photon transport is used to model three dimensional light scattering within a snow pack. NERD

LEDs are modeled as photon emitters according to their placement within the dome. An array of photons with wavelengths generated at random using a Gaussian distribution are used to mimic the 85 and 130 nm full width at half-maximum spectral emission characteristics of the narrow-band LEDs. Photons are initiated downward into the snow medium (Kaempfer et al., 2007) and propagated in optical depth space. Photon particle interactions are determined using random number generators and photons can either be absorbed or scattered with the probability determined by the particle single scatter albedo. Photons are

terminated upon absorption and followed if scattered. When a photon is scattered, its new direction cosines are determined by the specific particle scattering phase function.

To generate theoretical calibration curves mapping snow BRFs to snow SSA, we run multiple simulations for various particle SSA ranging from 10 to 90 m$^2$kg$^{-1}$. At least 100 thousand photons per simulation are propagated and followed through the snow medium until they are absorbed or exit the medium. The snow medium is modeled as a homogenous matrix of suspended

particles with input data containing the particle mass absorption cross section, asymmetry parameter, single scattering albedo, projected area, volume, and scattering matrix from Yang et al. (2013). Ice particle shape habits include spheres, droxtals, solid hexagonal columns, and solid hexagonal plates. We select these subset of shape habits from the larger dataset provided by Yang et al. (2013) because they are purely convex solid ice particles. Because they are convex bodies, their SSAs can be computed from the projected area and volume. To generate theoretical calibration curves mapping snow BRFs to snow SSA, we run

multiple simulations for various particle SSA ranging from 10 to 90 m$^2$kg$^{-1}$.

Reflected light from Lambertian surfaces is simulated using the Monte Carlo model to test its statistical uncertainty. To this end, azimuthal mean BRFs are calculated according to the reflectance definitions presented by Dumont et al. (2010) Hudson





et al. (2006), and F.E. Nicodemus et al. (1977). Accordingly, photon exit angles are grouped into 30 exit zenith angle ($\theta_r$) bins at three degree resolution. Azimuthal mean BRFs are calculated by zenith angle $\theta_r$ from the total incident photon flux $\Phi_i$ by

$$R(\theta_i; \theta_r) = \int\limits_0^{2\pi} \frac{d\Phi_r}{2\sin\theta_r \cos\theta_r \Phi_i} d\phi_r \qquad (5)$$

where $\Phi_r$ represents the azimuthally integrated photon flux through each $\theta_r$ bin. In the denominator, the $\cos\theta_r$ factor satisfies
Lambert's cosine law while $\sin\theta_r$ accounts for the zenith angular dependence of the azimuthally integrated projected solid angle. Finally, the factor two is necessary to normalize the resulting weighting function $w(\theta_r) = \sin\theta_r \cos\theta_r$, as

$$\int\limits_0^{\pi/2} \sin\theta_r \cos\theta_r d\theta_r = \frac{1}{2}. \qquad (6)$$

Equation (5) is applied to Monte Carlo simulations of 75 thousand photons reflected by Lambertian surfaces having reflectances of zero to one. At three degree resolution, 30 and 60 degree BRFs of Lambertian surfaces are simulated accurately to within +/-
2 %. Monte Carlo noise from 75 thousand photons are quantified by computing RMS errors across the full range of Lambertian reflectances. Across this range, RMS errors at 30 and 60 degrees are generally less than 0.01. These relatively small RMS errors computed from just 75 thousand simulated photons justify computing accurate BRFs at three degree resolution.

## 3 Results and Discussion

To examine the relationship between snow SSA and 1.30 and 1.55 µm BRFs, we compare X-CT derived snow SSA with
NERD snow measurements. To this end, we conduct side-by-side X-CT and NERD analysis of all snow samples described in the preceding section. In general, NERD BRFs are directly related to snow SSA (Fig. 4). At 1.30 µm, NERD snow BRFs are slightly higher at 60 degrees than at 30 degrees. Despite the direct relationships between NERD snow BRFs and X-CT derived snow SSA, there exists considerable spread in measurements at both wavelengths and at both viewing angles. The spread in measurements results in considerable uncertainty in the ability to retrieve snow SSA from NERD BRFs. In the following
subsections, we discuss NERD reflectance measurement validation and results from Monte Carlo simulations in the context of previous studies. Finally, we synthesize our findings in a subsection that gives an analytical calibration function relating NERD BRFs to snow SSA and discuss measurement uncertainty.

### 3.1 Reflectance Measurement Validation

Using the NERD, we can obtain relatively accurate snow BRF measurements in nature without drastically affecting the snow.
By recording measurements across two view azimuth angles and additional scattering planes by rotating the dome, we can assess azimuthal anisotropy in just a few minutes. Furthermore, by measuring multiple BRFs across multiple locations of a snow surface, we obtain numerous samples spanning multiple azimuthal planes that also enables easy characterization of the spatial variability in snow BRFs. Repeating rapid measurements in this manner allow us to obtain relatively accurate snow





BRFs. Multiple precise measurements allow quantifying relatively large BRF variations associated with azimuthal anisotropy and spatial heterogeneity. Median BRFs reported across a unique wavelength, LED position, and photodiode zenith angle give a second order approximation of the snow azimuthal mean BRF. Computing RMS errors from these uniquely defined wavelength-BRF combinations quantifies measurement uncertainty. To this end, we first test NERD accuracy, precision, and

responsivity by testing with idealized Lambertian surfaces before obtaining snow BRFs. Results in Table 1 indicate that any single NERD reading is subject to measurement uncertainty of about +/-2 %. Although measurement uncertainty prevents us from using the NERD to obtain highly accurate BRFs, NERD BRF measurements are accurate and precise enough to observe relatively large variations in snow BRFs that are of particular interest in this study.

Compared to the Infrasnow (Gergely et al., 2014), NERD BRF measurements of Lambertian surfaces are slightly less accu-

rate. In a similar validation experiment, Gergely et al. (2014) measure the reflectance of 0.25, 0.50, and 0.99 reflectance standards accurately to within less than 1 %. Gergely et al. (2014) use an integrating sphere that enables directional-hemispherical reflectance factor measurements at 950 nm in contrast to the 1.30 and 1.55 µm BRFs measured by the NERD. Both instruments make use of Lambertian reflectance standards for calibration and testing. Although each instrument uses a different wavelength and measures a different type of reflectance factor, testing on Lambertian reflectance standards with constant bidirectional re-

flectance distribution functions (BRDFs) allows for easy comparison of measurement uncertainty across multiple measurement techniques.

Dumont et al. (2010), for example, use Lambertian reflectors to report a BRF measurement accuracy of better than 1 % using a high angular resolution spectrogonioradiometer. Gallet et al. (2009) also use similar Lambertian standards to calibrate 1.31 and 1.55 µm directional-hemispherical reflectance factor measurements. Gallet et al. (2009) use six standards to paramet-

rically fit signal voltages to reflectance values. This approach accounts for nonlinear responsivity due to re-illumination of the standards through multiple scattering within the integrating sphere. While NERD responsivity is not perfectly linear, we expect re-illumination of the surface through multiple scattering within the black dome to be minimal.

Although photodiode responsivity varies with temperature, frequent calibration minimizes these errors. Therefore, the main source of NERD responsivity error is likely due to small deviations in light output from the LEDs. Like almost all electronic

circuit elements, LED performance is also a function of its temperature. In its operational mode, the NERD drives the user selected active LED with a current pulse width of two seconds. When the duty cycle is increased to 50 % (two seconds on, two seconds off), we observe drift in the photodiode response. This responsivity drift is mitigated, but not completely eliminated, when the duty cycle is decreased to 20 % (two seconds on, eight seconds off). Because we observe these responsivity errors in testing shortly after calibration, we speculate that changing LED temperatures can affect the the light output enough to cause a

one to two percent measurement error.

## 3.2   Monte Carlo Results

At 1.30 µm, 30 degree snow BRFs measured with the NERD for various snow SSA fall within the envelope of shape habits derived from Monte Carlo simulations. Monte Carlo simulations of spheres, droxtals, and hexagonal columns accurately predict 30 degree BRFs measured by the NERD for snow SSA ranging from 10 to 70 $m^2 kg^{-1}$. Monte Carlo simulations predict lower





BRF values at 60 degrees than at 30 degrees. These results provide an estimate of the uncertainty associated with deriving snow SSA from NERD BRFs across various shape habits and snow samples.

At 1.55 µm snow SSA values ranging from 10 to 70 $m^2kg^{-1}$ yield lower Monte Carlo simulated BRFs than what is measured by the NERD. Comparing 30 and 60 degree viewing zenith angles, Monte Carlo results are more similar at 1.55 µm than at 1.30
5 µm. The relationships between 1.55 µm BRFs and snow SSA are also more linear than those at 1.30 µm. Stronger linearity at 1.55 µm, however, does not necessarily imply more accurate snow SSA retrieval. Obtaining snow SSA at 1.55 µm is more difficult due to the lesser span and lower responsivity of snow BRFs at this wavelength.

### 3.3 NERD Snow SSA Calibration

In general, snow SSA results from X-CT scans are related to NERD 1.30 µm nadir illuminated BRFs via an exponential
10 relationship. This relationship exists at both the 30 and 60 degree viewing zenith angles. At 1.55 µm, snow SSA results from X-CT scans are related to NERD 15 degree, off nadir illuminated BRFs via linear relationships. The relationship between snow SSA and NERD measurements is most clear and robust at 1.30 µm. Nadir illumination at 1.30 µm results in the best snow SSA agreement across NERD observations and Monte Carlo modeling at the 30 degree viewing angle.

Our finding of the exponential relationship between snow SSA and 1.30 µm BRFs is consistent with previous studies (Pi-
15 card et al., 2009; Gallet et al., 2009). Likewise, Gallet et al. (2009) identify a linear relationship between 1.55 µm reflectance and snow SSA. These studies, however, quantify snow SSA from hemispherical reflectances instead of BRFs. Hemispherical reflectance measurements theoretically reduce measurement variations associated with grain shapes. Picard et al. (2009) conclude that obtaining snow SSA from snow albedo measurements are subject to as much as 20 percent error when grain shape is unknown. This relatively large source of error due to grain shape is further explored here in Monte Carlo derived albedo
20 calculations for snow surface of spheres, droxtals, solid hexagonal columns, and hexagonal plates (Fig. 5).

As expected, snow modeled as spherical ice particles, simulated in the Monte Carlo model using the Henyey-Greenstein phase function

$$P_{\mathrm{HG}}(\cos\theta; g) = \frac{1 - g^2}{(1 + g^2 - 2g\cos\theta)^{3/2}}, \tag{7}$$

where $\theta$ is the scattering angle and $g$ is the relevant asymmetry parameter, most closely agrees with 1.30 and 1.55 µm narrow
25 band black-sky snow albedo calculated from the Snow, Ice, and Aerosol Radiation (SNICAR) model (Flanner et al., 2007). Snow albedo dependence on grain shape is consistent at both wavelengths. In general, droxtals yield higher reflectances. Reflectances of solid hexagonal columns agree closely with spheres and SNICAR at both wavelengths for snow SSA lower than 40 $m^2kg^{-1}$, after which they tend toward reflectances similar to droxtals. Finally, hexagonal plates yield low reflectances. Low reflectances at both wavelengths are due to the extremely sharp forward scattering peak of these plates. Although highly
30 idealized and perfectly smooth, these shape habits demonstrate the relatively large hemispherical reflectance variations across snow grain shape. These large variations in reflectance across grain shape are the largest source of uncertainty in snow SSA measurements using infrared reflectance. Monte Carlo modeling of BRFs in Fig. 4 also suggest these uncertainties exist for





directional reflectance measurements. These uncertainties associated with unknown grain shape limit accuracy of NERD SSA retrieval.

Surprisingly, 1.55 µm BRFs measured by the NERD are higher than predicted by Monte Carlo modeling. Using the NERD, we observe 1.55 µm snow BRFs as high as 0.2. We measure the highest 1.55 µm snow BRFs at 60 degrees for particularly
fresh snow, but high 1.55 µm BRFs are larger than simulated for all SSA. Because of its relatively high instrument precision (Table 1), these seemingly high BRFs are probably accurate. The primary contributor for the discrepancies against models at this wavelength is possibly due to the broad spectral emission characteristics of the 1.55 µm LEDs. With full width at half maximums of 130 nm, non-negligible light emission at wavelengths much shorter, toward the near-infrared, is a likely cause of higher than expected reflectances. Although the spectral emission characteristics of NERD LEDs are simulated in Monte Carlo
simulations using Gaussian photon wavelength distributions, and in SNICAR using a simple normalized Gaussian weighting function, non-negligible light emission from the tails of these distributions is possibly under estimated. Because of the expected sharp increase in snow reflectance as wavelength decreases from 1.55 to 1.30 µm (Wiscombe and Warren, 1980; Flanner et al., 2007), it is possible that even a small amount of light emission at wavelengths toward the near-infrared can have a measurable effect on snow BRF observations. This effect is further explored in Monte Carlo simulations by broadening the Gaussian
distribution of photon wavelengths and in SNICAR by broadening the Gaussian weighting function applied to narrow-band albedo calculations. These calculations confirm this hypothesis, as 1.55 µm narrow band albedo with a full width at half maximums of 0.26 µm (doubled from 0.13 µm) closely agree with NERD BRF measurements. This finding suggests light emission from the 1.55 µm LEDs is non-negligible at shorter, more absorptive wavelengths.

Notwithstanding the limitations associated with retrieving precise snow SSA from BRFs, we generate an analytical cali-
bration function relating snow SSA to NERD BRFs. To this end, we propose the general exponential form for 1.30 µm snow BRFs, such that

$$\text{SSA} = \alpha \exp(R_{1.30}) + \beta \tag{8}$$

for predicted snow SSA and 1.30 µm snow BRF $R_{1.30}$. Using least squares regression analysis, we compute parameters $\alpha$ and $\beta$ for both 30 and 60 degree viewing zenith angles. At 30 degrees, setting $\alpha = 88.7$ and $\beta = -103$ minimizes residuals and results
in a snow SSA RMS error of 7.05 $m^2kg^{-1}$ (Fig. 6, left). At 60 degrees, setting $\alpha = 91.7$ and $\beta = -113$ minimizes residuals and results in a snow SSA RMS error of 7.23 $m^2kg^{-1}$ (Fig. 6, right).

This margin of uncertainty regarding SSA retrieval from snow infrared reflectance measurements falls within the expected range reported in previous studies (Picard et al., 2009; Gallet et al., 2009). This analysis complements previous studies and indicates that retrieval of highly precise snow SSA using NERD measurements is unlikely. Obtaining approximate estimates
of snow SSA using NERD measurements across a wide variety of snow types, however, is highly likely. Because of its non-destructive nature, rapid, repeatable retrieval of approximate snow SSA using the NERD will be useful for studying hourly-scale snow metamorphosis (Fig. 7). While the 1.30 µm, 30 degree viewing zenith angle BRF combination most closely agrees with modeled BRFs, a similar margin of error at the 60 degree viewing zenith angle can provide a second estimate of snow



SSA. Reporting two snow SSA values using both view angles can ultimately give observationalists an idea of the variability in SSA retrieval resulting from the angular dependence of the snow BRDF in the near-infrared.

While these results minimize the usefulness of obtaining snow SSA from 1.55 µm snow BRFs, it is worth noting that Gallet et al. (2009) use 1.55 µm in their DUFISSS to obtain measurements of large SSA snow ($> 60$ m$^2$kg$^{-1}$). Here, nearly all snow samples used in the NERD SSA calibration were lower than this threshold. A possible follow on study would include snow of higher SSA to determine the utility of 1.55 µm snow BRFs in measuring fresh snow of extremely high SSA particularly common in the extremely cold Arctic and Antarctic environments as in Legagneux et al. (2002) and Libois et al. (2015).

## 4 Conclusions

To obtain quick, accurate, reliable, and repeatable measurements of snow SSA, we engineered an instrument that measures snow 1.30 and 1.55 µm BRFs. By flashing narrow band LEDs centered around these wavelengths, light reflected by experimental snow surfaces is measured using photodiodes mounted at 30 and 60 degrees relative to nadir. Photodiode currents are converted into measurable voltage signals enabling calibrated BRF calculations using Lambertian reflectance targets. Monte Carlo modeling and X-CT derived snow SSA help to demonstrate the relationship between snow BRFs and SSA. Generally, we found an exponential relationship between 1.30 µm BRFs and snow SSA. These results demonstrate the NERD's ability to obtain estimates of snow SSA to within +/- 7 m$^2$kg$^{-1}$ without destroying snow samples. This nondestructive technique for snow SSA retrieval will be useful in science applications that involve hourly scale monitoring of snow SSA. Applying the NERD will be especially useful in experiments designed to learn about the effects of LAI on snow metamorphosis and to explore the spatial heterogeneity of snow SSA. Because it can operate quickly, NERD measurements will also complement satellite borne observations during narrow sampling windows.

*Code and data availability.* Plot data referenced in this manuscript and associated Python scripts used to generate figures are made available via the University of Michigan's Deep Blue data repository (Schneider, 2018).

## Appendix A

The objective of this appendix is to show that for convex bodies, sphere effective radii $r_{eff}$, as defined in Eq. (2), and RE, as defined in Eq. (3), are equivalent. In Vouk (1948), it is shown that for convex bodies,

$$S = 4\bar{A}, \tag{A1}$$

where $\bar{A}$ is the average projected area of the convex body. Substituting Eq. (A1) into Eq. (1) then gives

$$\text{SSA} = \frac{4\bar{A}}{\rho_{ice}V}. \tag{A2}$$



Equating Eq. (2) and Eq. (A2) and simplifying,

$$3/r_{eff} = \frac{4\bar{A}}{V}. \tag{A3}$$

Finally, solving Eq. (A3) for $r_{eff}$ gives

$$r_{eff} = \frac{3}{4}(V/\bar{A}), \tag{A4}$$

5 which is equivalent to the expression for RE given in Eq. (3), thus concluding the proof.

*Competing interests.* We are not aware of any competing interests associated with the publication of this manuscript.

*Acknowledgements.* This work is funded, in part, by the National Science Foundation, grant number ARC-1253154.

The authors would like to thank colleagues at the Cold Regions Research and Engineering Laboratory (CRREL) in Hanover, New Hampshire for their generous support. In particular, thanks to Zoe Courville and John Fegyveresi for their hospitality and guidance navigating the

10 facilities at CRREL. We also thank Ross Lieblappen for sharing his micro-computed tomography expertise through providing a thorough tutorial for running and analyzing snow scans. Finally, thanks to Alden Adolph for facilitating travel accommodations and welcoming Adam to Hanover, New Hampshire in 2016 and 2017.





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





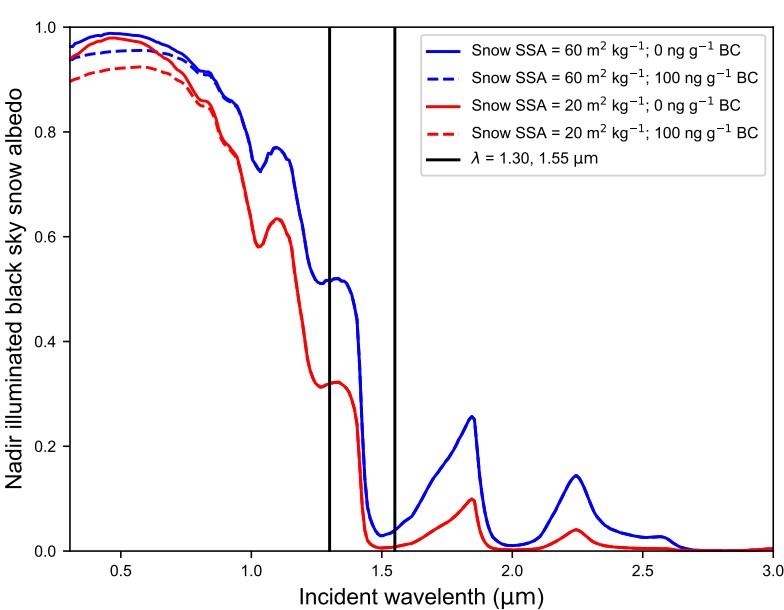

**Figure 1.** Black sky spectral snow albedo under nadir illumination simulated using the Snow, Ice, and Aerosol Radiation (SNICAR) model (Flanner et al., 2007). Solid curves represent clean snow of medium-high SSA ($60 \ m^2 \ kg^{-1}$, blue) and medium-low SSA ($20 \ m^2 \ kg^{-1}$, red) to show the dependence of snow albedo on snow SSA. Dashed lines represent contaminated snow with uncoated black carbon (BC) particulate concentrations of $100 \ ng \ g^{-1}$. $100 \ ng \ g^{-1}$ of BC in snow directly reduces visible but not infrared albedo. The dependence of snow albedo on SSA but not on BC concentration at 1.30 and 1.55µm motivates the use of these wavelengths for measurement of snow grain size.



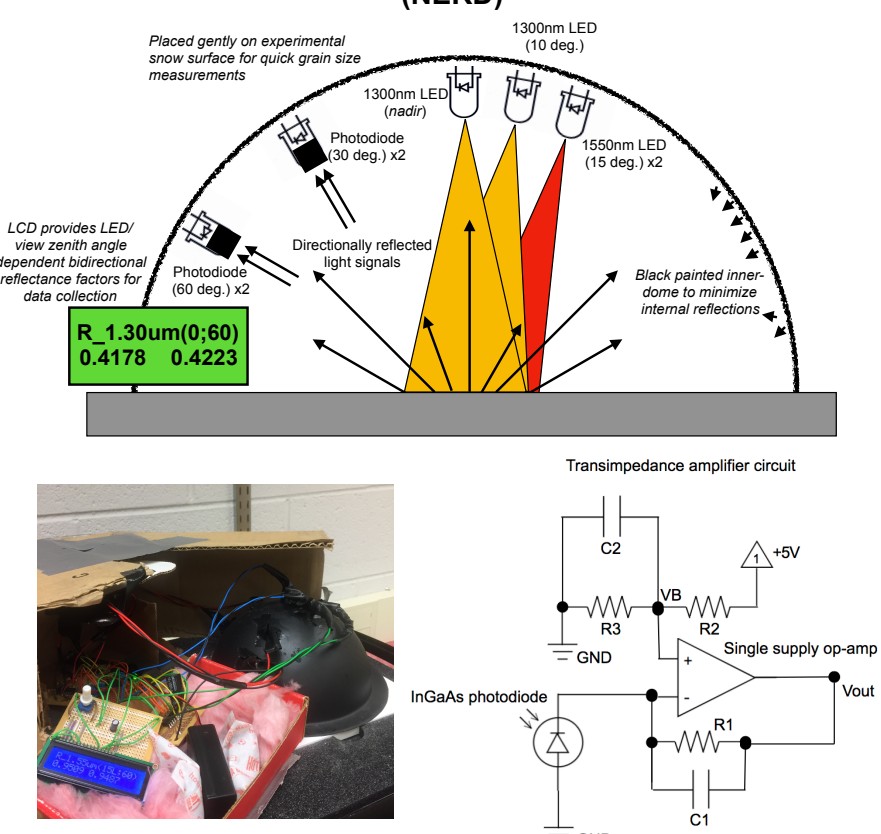

**Figure 2.** Near-Infrared Emitting and Reflectance-Monitoring Dome (NERD) schematic, photograph, and transimpedance amplifier circuit diagram. Two instruments are engineered with different photodiode responsivities. Photodiode responsivities are determined by the feedback resistance (R1) in the transimpedance amplifier circuits. Using feedback resistances of as low as four mega-Ohms in a low responsivity NERD and as high as 15 mega-Ohms in a high responsivity NERD yield dynamic reflectance factor responses over the range of 0 to 0.95 at 1.30 and 1.55 μm.





**Figure 3.** X-ray microcomputed tomography (X-CT) images of snow samples (15 mm diameter) collected across three winters (2015-2017) in Hanover, New Hampshire. Snow samples shown on the left (a., b., c.) were scanned during 2016, while those on the right (d., e., f.) were scanned in 2017. Generally (except for b.), snow specific surface area, derived from X-CT analysis software, decreases as snow grains appear more rounded.





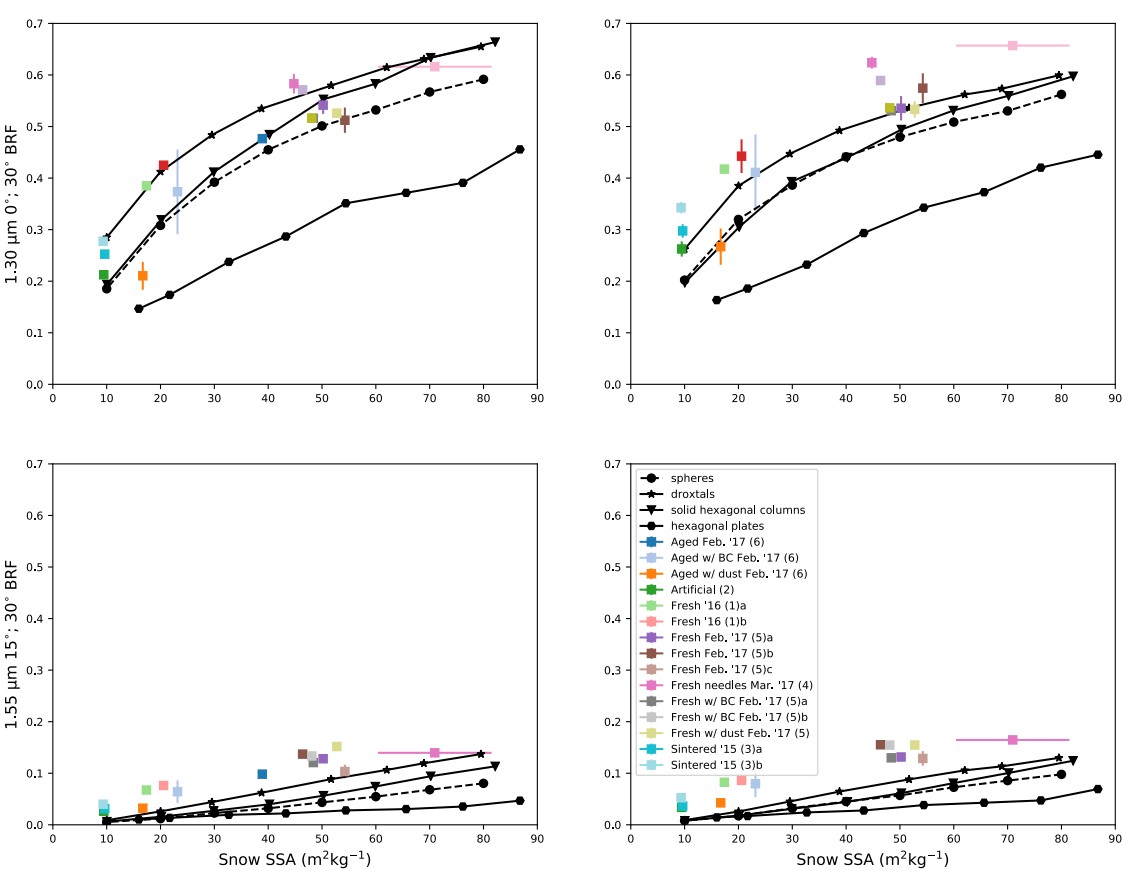

**Figure 4.** 1.30 (top) and 1.55 (bottom) μm 30 (left) and 60 (right) degree bidirectional reflectance factors (BRFs) versus snow specific surface area (SSA). Black line segments connect BRFs calculated from Monte Carlo simulations of photon pathways through snow mediums comprised of spheres (circles, dashed lines), droxtals (stars), solid hexagonal columns (triangles), and hexagonal plates (hexagons). Measured BRFs with the NERD are scattered against X-CT derived snow SSA (colored squares). Snow samples labeled in the key relate directly to those described in the previous section. Vertical error bars on NERD BRFs represent standard deviations calculated from multiple azimuthal samples. Horizontal error bars on X-CT derived SSA, where present, represent standard deviations from multiple scans on similar snow samples.





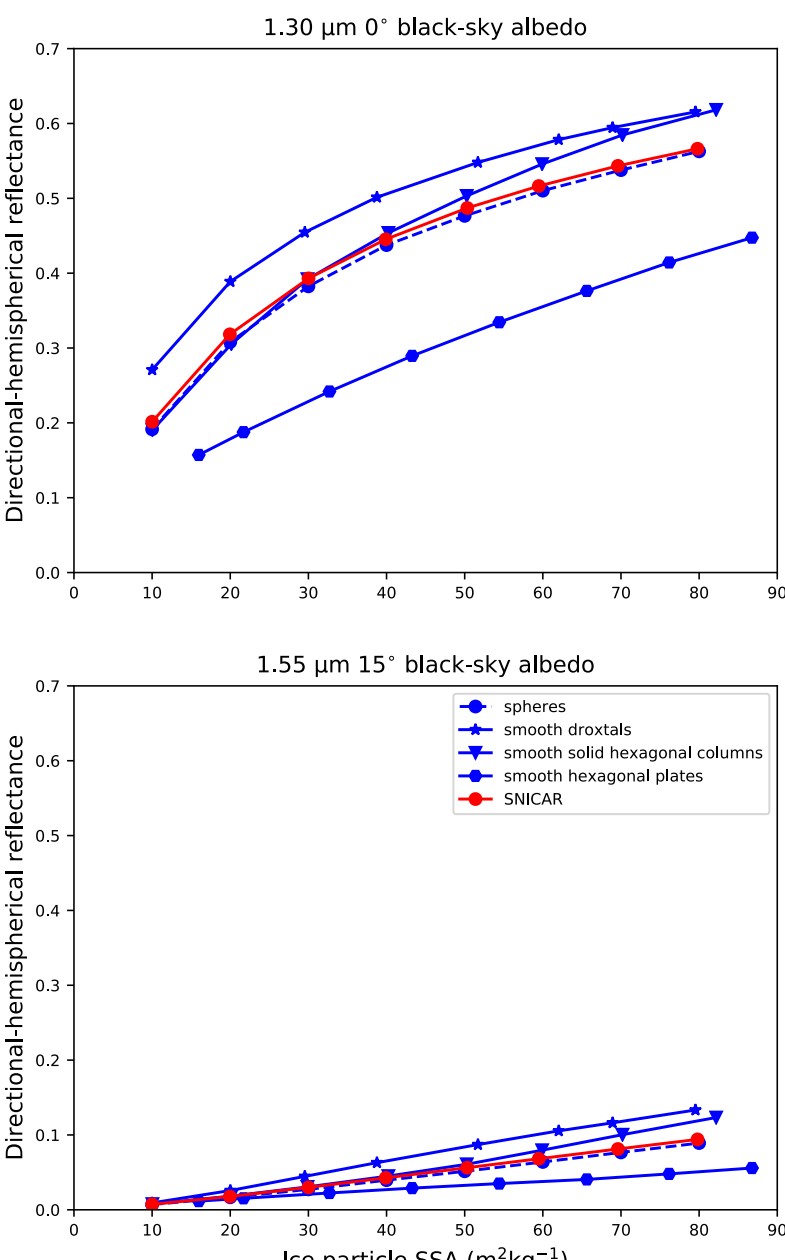

**Figure 5.** Modeled 1.30 µm nadir (top) and 1.55 µm 15 degree (bottom) directional-hemispherical reflectance for various snow SSA. Blue line segments connect albedo calculations from Monte Carlo simulations of photon pathways through snow mediums of spheres (circles, dashed lines), droxtals (stars), solid hexagonal columns (triangles), and hexagonal plates (hexagons). Red line segments connect albedo calculations from the Snow, Ice, and Aerosol Radiation (SNICAR) model (Flanner et al., 2007).





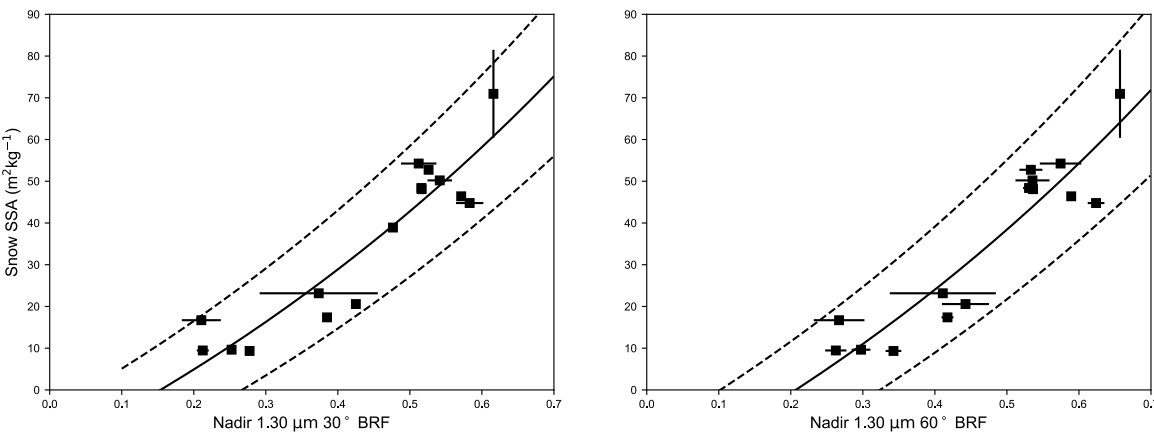

**Figure 6.** Near-Infrared Emitting and Reflectance-Monitoring Dome (NERD) snow specific surface area (SSA) calibration. Markers represent scattered X-CT derived snow SSA against nadir 1.30 μm 30 (left) and 60 (right) degree bidirectional reflectance factors (BRFs) measured by the NERD (also plotted in Fig. 4). Curves show center (solid), top and bottom (dashed) estimates of the analytical expression in equation 8. These are calculated from three $\alpha$ parameters (88.7+/- 9.50 $m^2 kg^{-1}$ at 30 degrees; 91.7+/- 10.13 $m^2 kg^{-1}$ at 60 degrees) using least squares regressions and their associated standard errors of the gradients (i.e., slopes).





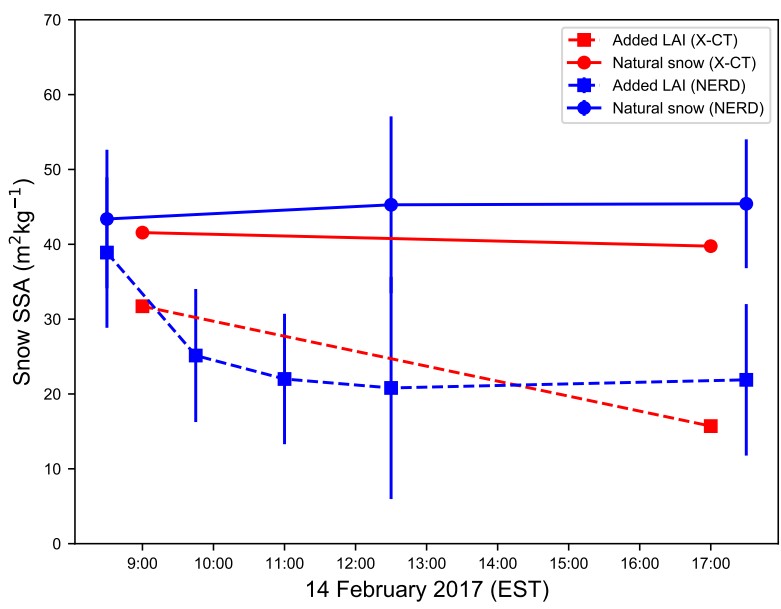

**Figure 7.** Snow specific surface area (SSA) measured throughout the day on 14 February 2017. Morning (09:00) and afternoon (17:00) samples were transported to the nearby Cold Regions Research Engineering Laboratory (CRREL) in Hanover, New Hampshire for X-ray microcomputed tomography (X-CT) analysis. SSA measurements derived from X-CT scans are shown in red. In blue, NERD SSA estimates derived from 1.30 μm 60 degree BRFs (Eq. (8)) depict hourly-scale snow metamorphosis. Dashed line segments connect evolving snow SSA estimates of snow samples with added dust to induce rapid snow metamorphosis. Vertical error bars on NERD SSA estimates represent margin of uncertainty associated with calibration error plus measurement standard deviations. These results contain the first measurement data obtained by the NERD used to determine snow SSA. Because of its nondestructive nature, this technique enables the study of snow metamorphosis in situ on hourly time scales.





**Table 1.** NERD Lambertian Reflectance Measurements

| $\lambda = 1.30$ μm | | Median BRF (RMS error) | | | |
|---|---|---|---|---|---|
| $n$ | $\rho_L$ | $R(0°; 30°)$ | $R(0°; 60°)$ | $R(10°; 30°)$ | $R(10°; 60°)$ |
| 10 | 0.422 | 0.399 (0.021) | 0.422 (0.016) | 0.415 (0.015) | 0.434 (0.015) |
| 10 | 0.951 | 0.939 (0.013) | 0.944 (0.015) | 0.958 (0.018) | 0.952 (0.010) |
| $N$ | | Linear regression; $\hat{R}(\rho_L) = A\rho_L + B$ | | | |
| 20 | $\hat{R} =$ | $\{1.023\rho_L$ - 0.028, | $0.987\rho_L$ + 0.007, | $1.031\rho_L$ - 0.024, | $0.980\rho_L$ - 0.018$\}$ |
| $\lambda = 1.55$ μm | | Median BRF (RMS error) | | | |
| $n$ | $\rho_L$ | $R(15_a^\circ; 30°)$ | $R(15_a^\circ; 60°)$ | $R(15_b^\circ; 30°)$ | $R(15_b^\circ; 60°)$ |
| 10 | 0.413 | 0.410 (0.009) | 0.420 (0.017) | 0.411 (0.008) | 0.420 (0.021) |
| 6 | 0.944 | 0.959 (0.012) | 0.963 (0.019) | 0.960 (0.013) | 0.964 (0.020) |
| $N$ | | Linear regression; $\hat{R}(\rho_L) = A\rho_L + B$ | | | |
| 16 | $\hat{R} =$ | $\{1.028\rho_L$ - 0.016, | $1.016\rho_L$ + 0.003, | $1.026\rho_L$ - 0.014, | $1.011\rho_L$ + 0.009$\}$ |