# Peer review of "Monitoring of Snow Surface Near-Infrared Bidirectional Reflectance Factors with Added Light Absorbing Particles"

_The Cryosphere, 2018_

## Referee Comment (RC1) · Dumont (Referee) · 28 Nov 2018

**Review of "Measuring Snow Specific Surface Area with 1,30 and 1,55 um Bidirectional Reflectance Factors" by Schneider et al.**

First of all, I am very sorry for the delay of my comments.

**Summary**
This paper presents an interesting new instrument to estimate SSA of the surface snow from NIR reflectance in a non destructive way. The instrument is described in detail and calibrated. The accuracy is evaluated using model results and X-ray tomography measurements.

**Recommendations**
Overall, I think this paper describes an interesting new instrument to measure surface SSA along with a thorough evaluation of the accuracy and limits. However, I think that major revisions are required before publication as detailed in my specific and minor comments below.

**Specific comments**

1/ The introduction is in my opinion, a bit too scattered and confusing and some literature references are also missing. More specifically,
a - it's a bit weird to have references to calculation in an appendix in the introduction. For me, either the calculation already exists in the literature and then it would be nice to add the reference or it's a new result that should be included in the results section
b – page 2, lines 22-31 , I don't think it's necessary to go into too much details about the SSA evolution in time, a few sentences without any equation should be sufficient. It's not directly related to the objective of the paper and would give more space for discussing the sate of the art of SSA measurements
c - page 3, lines 1-8, this section is really important for the rest of the paper. It seems to me that it would worth more details on the methods (advantages and drawbacks) and accuracy. Several methodologies are missing here such as IR photography (Matzl and Schneebeli, 2006) , SMP (Proksch et al. 2015) , and retrieval from spectral albedo which is also non destructive (Picard et al., 2016 , Dumont et al., 2017).
Regarding SSA calculation from X-ray imaging, I think adding some discussion also on the methodology and resolution issue would be nice (e.g. Hagenmuller et al., 2016).

d- Since you also present a Monte-Carlo model, maybe a short state of the art of existing theory and models to simulation snow BRF need to be added and why a new Monte-Carlo model is required ?
e.g. Malinka, 2014 , Kokhanovsky and Zege, 2004 , Xi et al., 2006 ….

2/ Section 2.1. Some details are missing here (but maybe I did not check carefully enough), what is the diameter of the illumination ? How homogeneous is it ? What is the FOV of the photodiode ? In which azimuthal planes are they with respect to the illumination ?

3/ Page 7, Equation 5. Here I probably misunderstood something, why is the BRF averaged over all azimuths while the measurement is done only in two azimuthal planes ?

4/ Section 3, I think it would be less confusing for the reader to start with the model evaluation first.

5/ Section 3.1. The section is long and a bit confused, can it be re-arranged ?

6/ Section 3.3, comparison with SNICAR should in my mind be part of the model evaluation. It's a bit confusing to have it mixed with the calibration.

7/ In the discussion, I would also add some details on the surface roughness effects and liquid water effect maybe (e.g. Gallet et al., 2014)

8/ To my mind, both the conclusions and the abstracts should more clearly state the advantages and drawbacks of this new instrument compared to existing ones. The estimated accuracy in the SSA measurements  should also be stated in the abstract.

**Minor comments**
Page 3, line 32, "flat black paint", it would be super interesting to know the spectrum, flat in which range ? I think these details are important for the discussions in the end of the paper.
Section 2.2. An accuracy assessment of the SSA calculation from the X-ray images would be nice. I think 14,6 microns is quite rough for snow types of snow (e.g. e and a in Fig. 3).

Section 2.3 Maybe a table would be clearer than a text description.

Page 6 line 19 scatter → scattering

Page 6 line 20. After how many scatter do you stop following the photon ?

Page 10 -  last section, Picard et al., 2016 and Dumont et al., 2017 provide a detail assesment of the SSA retrieval uncertainties.

Page 11 – lines 3-7, this should be also indicated in the introduction.

---

## Referee Comment (RC2) · Anonymous Referee #2 · 2 Dec 2018

Review of "Measuring snow specific surface area with 1.30 and 1.55 µm bidirectional reflectance factors" by Schneider et al.

The manuscript presents an interesting device to retrieve snow SSA using active optical sensors. However, the language is not appropriate and requires an extensive revision before publication, the paper is badly organized, and the analysis of the results needs to be deepened. I therefore recommend a major revision.

Main problems:

1. The introduction is in some parts confused and not focusing on the main message. It needs to describe the state of art of the snow SSA measurements (now this is condensed in a single paragraph), and possibly the issues that make the SSA measurements so challenging.
2. The paper includes too many unnecessary technical details. The description of technical details can be justified only if it serves the demonstration or clarification of concepts. This is scientific paper, not a technical manual for users.
3. Results and discussion section need to be reorganized and rewritten. First, a thorough presentation of the results needs to be made (now it is too superficial), and, only after that, a discussion on the usability of NERD can be made (now it is at the beginning of sect 3.1, i.e. before the description of the modelling results!). Instead of discussing the opportunity of using reflectance standards to calibrate SSA detectors based on active optical sensors (which is a very common and obvious procedure), the authors could focus on the implications of the differences in the available devices, and highlight their strengths/weaknesses.

Detailed comments:

p.1, L17-18: "Positive snow internal albedo feedback occurs due to the strong dependence of snow infrared reflectance on snow specific surface area (SSA)." This sentence is too compact. Please explain this internal albedo feedback more explicitly.

P1., L18-20: "The Snow, Ice, and … in Fig. 1". Fig 1 is not sufficiently justified here. It should be moved later in the paper, when describing the reason for the selection of the wavelengths 1300 and 1550 nm for the detection of SSA.

p.2, L9: "…are equivalent for convex bodies (see Appendix A)." There is no need to write an appendix to make a geometrical demonstration that was already derived more than 150 years ago. Instead, please refer to some book of convex geometry, or better to the original demonstration by Cauchy (as done in Pirazzini et al.: "Measurements and modelling of snow particle size and shortwave infrared albedo over a melting Antarctic ice sheet", The Cryosphere, 9, 2357-2381, https://doi.org/10.5194/tc-9-2357-2015, 2015).

p.2, L16-17: "observe seasonal scale snow albedo decline in springtime Colorado". Could you please improve the expression, for instance as "observe snow albedo decline during the spring season in Colorado"?

p.2, L17: "In contrast, however, they find that snow albedo is primarily related to dust concentration." This sentence is incorrect. First of all, the snow albedo is mostly determined by the optical properties of the snow, and not by dust concentration. You may want to say that it is affected by dust concentration, but you cannot claim that it is the main albedo driver. Secondly, why you wrote "In contrast"? In the paper by Skiles and

Painter (2017) the springtime albedo decline was accelerated by the dust load, which concentrated at the surface during the progress of the melting further decreasing the albedo. Hence, the increase in dust concentration at the surface affected the observed albedo decline, and was not in contrast with it.

p.2, L19: "where the albedo reduction…" Instead of "where" I suppose you meant something like "who showed that…", right?

p.2, L21: "snow internal albedo feedback" shouldn't be "internal snow albedo feedback"? As pointed out in my comment above, it is not at all clear what you mean for "internal" snow albedo feedback. Please explain.

p.2, L23-24: "Surface warming can also reduce snow grain growth rates, however, if growth processes from vapor diffusion and strong temperature gradients are affected negatively (Flanner and Zender, 2006)." The meaning of this sentence is very obscure. Could you explain more clearly what you mean, without requiring from the reader to study Flenner and Zender in order to understand what you mean?

p.2, L25-31: "Recent studies …" This section seems to be out of context: it is not linked to the purpose of the paper. Please remove it, or explicitly explain the connection with the content of the paper.

p.3, L16: "The NERD is designed to measure 1.30 and 1.55 µm BRFs". Please explain here why these wavelengths were selected, and highlight here the analogy with DUFISSS in the wavelength selection.

p.3, L25: "The NERD is similar to that of … in that it uses …." Please reformulate the sentence improving the linguistic expression and moving it above (see previous comment).

p.3, L27-30: "LEDs are toggled … (20% duty cycle)" A lot of not needed technical details. Please remove.

p.3, L31-32: "Here, rather, we direct photodiodes toward the illuminated surface in a black dome to measure BRFs" The linguistic expression is particularly poor in this sentence. Instead of "we" use a passive expression.

p.4, Sect 2.1 and 2.2. Please remove all the technical details that do not provide any added value to the interpretation of the measurements. E.g. "Waiting 0.75 seconds after toggling the LED allows for enough time for the photodiode current to stabilize. After these currents stabilize, 100 voltage samples (ranging from 0.1 to 1.0 Volts) are then rapidly collected using the Ruggeduino-ET's ADCs. The average voltage obtained during active illumination is differenced from the average dark current voltage to derive reflectance 10 factors.", "The reflectance of the targets are measured with high precision across a broad spectrum. At 1.30 (1.55) µm, the white and gray targets have calibrated reflectances of 0.95073 (0.94426) and 0.42170 (0.41343), respectively, as reported by the manufacturer.", "Small samples of snow are collected in roughly 10 cm tall cylindrical plastic sample holders and placed into the machine. An X-ray source is emitted at 40-45 kV and 177-200 micro-Amps. X-ray transmittance is measured as the machine rotates the sample. Setting the exposure time to 340 ms at a pixel resolution of 14.9 µm with rotation steps at 0.3-0.4 degrees allow for fast scan times of roughly 15 minutes. These short scan times are necessary to complete the scan without too much absorbed radiation melting the snow."

p.4, L19: "Using both…" What do you mean for "both"?

p.5, Sect 2.3. This section needs to be rewritten in a much more compact and consistent way. Expressions such as "oldest class" are meaningless. You should really apply the snow descriptors listed in The International Classification for Seasonal Snow on the Ground (Fierz, 2009). Instead of repeating 6 times that the

measurements were performed in Hanover and samples were transported to CRREL for X-ray microTomography, focus on the characteristics of the different samples. Eliminate subparagraphs and unnecessary details such as "…distinguishable only by the container they were stored in…", or the sentences in lines 24-27 (until "…nearby lab for X-CT analysis"), and the not relevant sentence "All samples with added LAI included in the NERD SSA calibration dataset were first screened to remove samples with heavy LAI loads that caused direct snow darkening at 1.30 5 and 1.55 μm."

p.6, Sect. 2.4. You need to provide some introduction explaining the purpose of the model simulations

p.6, L30: replace "multiple" with "BRFs"

p.7, L24 and following: this sections need to be moved after the thorough presentation of the results.

p.8, L12-13: "Both instruments make use of Lambertian reflectance standards for calibration and testing." This is a repetition: you have already explained in the lines above. Please remove it. Instead of discussing the opportunity of using reflectance standards to calibrate SSA detector based on active optical sensors, you could focus the discussion on comparing the different working principles, and strengths/weaknesses of the devices.

p.8, L13-16: "Although each instrument…" This sentence is a rather obvious statement that does not add anything to the paper. Please remove.

p.8, L23: "Although photodiode responsivity varies with temperature, frequent calibration minimizes these errors" This is a critical point that deserves further explanation. Is calibration required on the field before/after each measurement (as done for instance when using the IceCube device)? If this is the case, please explain, and describe the needed measurement procedure, including calibration.

p.8, L34 – p.9, L1: "Monte Carlo simulations predict lower BRF values at 60 degrees than at 30 degrees". This sentence refers to radiances at 1.30μm: looking at Fig. 4 I see the opposite, i.e. that for most grain shapes, when SSA is larger than 40 $m^2kg^{-1}$ BRF is larger at 60 degrees than at 30 degrees.

p.9, L17: "Hemispherical reflectance measurements theoretically reduce measurement variations associated with grain shapes". Why? Please explain. Comparing Fig. 4 (top) and Fig. 5 (top) where, respectively, BRFs and directional-hemispherical albedo are illustrated, I would say that both hemispherical and directional measurements show a very similar dependence on grain shape. In my opinion, Fig. 4 would deserve a much deeper analysis. For instance, why the BRFs measured with NERD are so much higher than the model results in 1.30 μm at 60 degrees? And why the modelled BRFs at 1.30 μm are lower at 60deg than at 30 deg? Etc…

p.9, L31-32: "These large variations in reflectance across grain shape are the largest source of uncertainty in snow SSA measurements using infrared reflectance." I disagree. Even larger uncertainties can be associated to the instrument set up in certain snow conditions. You have not discusses the effect of natural light entering into the dome and detected by the photodetectors. Probably, you will have this unwanted light source every time the target snow surface is not perfectly smooth, unless you insert the edges of the dome for several millimeters inside the snow surface. With other optical-based devices to derive SSA (such as DUFISSS and IceCube), a large source of uncertainty may derive from the snow sampling procedure (especially in case of surface hoar or very soft new snow). In my opinion, even your Fig. 4 shows that the large scatter in the optically derived SSA is not only attributable to a grain shape effect. The instrumental and set up error sources deserve much more discussion.

p.10, L16-17: "These calculations confirm this hypothesis, as 1.55 μm narrow band albedo with a full width at half maximums of 0.26 μm (doubled from 0.13 μm) closely agree with NERD BRF measurements." Please show these results in a Figure.

Figure 2: A much clearer photo of the sensor is needed, which would show only the essential components. The text in the figure should be less technical, or the technical terminology should be explained (what is the meaning of "LCD"? Is the whole sentence "LCD provides … data collection" needed? If yes, you should better explain its content, possibly in the main text and not in the figure. Is the diagram of the Transimpedance amplifier circuit needed? Instead of providing so many technical details, you should explain what the achieved performance is and why it is needed. Also the meaning and scope of the sentence "Using feedback resistances as low as …" in the figure caption is totally obscure. What is the scientific message behind it?

Figure 4: Please mark the vertical and horizontal grids, to facilitate the comparison among the plots.

Figure 5: what is the added value of this figure? The considerations on the effect of grain shape drown on the basis of directional-hemispherical albedo calculation can equally well been drown on the basis of Fig 4 (showing BRF calculations). I would simply remove the figure.

Table 1: in the table caption please explain the meaning of the used symbols and the content of each column and row.

---

## Referee Comment (RC3) · Anonymous Referee #3 · 5 Dec 2018

Review:
Measuring snow specific surface area with 1.30 and 1.55 um bidirectional reflectance factors
A. Schneider, M. Flanner, and R. De Roo

Summary:

The authors present a novel empirical measurement method to derive snow specific surface area (SSA) from reflectance measurements that may be used to develop a tool for snow SSA measurements in the field, called the NERD. The impact of some snow physical characteristics on measured bidirectional reflectance factors (BRFs) is evaluated, but eventually the SSA measurement method is formulated by resorting to an empirical relationship between measured snow BRF and snow SSA determined by computed tomography (CT) for a small number of samples, without providing a quantitative uncertainty assessment or validation of the SSA measurement method.

While the manuscript also contains some interesting mostly qualitative results beyond the presentation of the measurement method, e.g., the influence of the 'snow grain shape' used for modeling BRFs and the preliminary results given in Fig. 7, several major points should be addressed before publication is considered. The most critical ones are: overly optimistic and premature claims about the presented measurement principle that are based on very limited analysis results; overly simplified and misleading generalizations about previously presented measurement methods that can also be used to derive snow SSA; there is no validation of the presented SSA measurement method against an independent measurement method and no quantitative discussion of SSA measurement uncertainties. The manuscript could also benefit from a more streamlined structure and a clearer writing style.

General comments:

1)
Validation of the NERD measurement principle would entail comparing snow SSA values measured with the NERD to independent SSA measurements, e.g., SSA measured by other optical measurement methods or by methane gas absorption (or at the very least a vast number of CT measurements that are not used for developing the NERD measurement method).

The authors have merely shown the consistency of BRF measurements (not SSA measurements) performed with the NERD on highly uniform, diffusely scattering reflectance standard surfaces. Neither the effects of extensive volume-scattering in snow nor possible directional/specular effects at the snow surface are included here. For example, is there light leakage into or out of the NERD on uneven surfaces, and what effect do oriented surface structures (small dents or ripples on the surface) have on measured BRF (I would expect this to be of greater concern here than for previously presented methods that are based on diffuse instead of directional reflectance)?

Furthermore, no systematic and quantitative uncertainty assessment of snow SSA measurements with the NERD is presented (although the impact of various sources of uncertainty on measured BRF is discussed), which is an integral part of any presentation of a novel measurement method, as described in the 'Guide to the expression of uncertainty in measurement', for example. Particularly an intercomparison of multiple measurement methods or comparisons of in situ and remote sensing observations, as alluded to in the conclusions of the manuscript, require a thorough assessment of the uncertainties affecting each measurement method to allow drawing reliable conclusions from such comparisons. In this manuscript, only a rough estimate of the variability of an empirically derived exponential relationship between BRF and SSA is interpreted as SSA measurement uncertainty, based on a low number of sample measurements. For empirical measurement methods,

i.e., measurement methods that are not based on physical measurement models but on statistical correlation as presented in this manuscript, a large number of samples is required to guarantee a high statistical significance. A low number of samples can easily lead to highly underestimated (or overestimated) measurement uncertainties over the entire or parts of the measurement range, especially when not validated against independent measurement methods.

A detailed NERD SSA measurement uncertainty assessment and validation would also include how the mismatch between CT measurement volume and NERD measurement volume affects the analysis. While suggesting to use the NERD to determine the large-scale spatial variability of snow SSA, the authors fail to discuss this very effect at a small scale of ~10 cm, relevant for the NERD measurement principle, e.g., how do CT samples and NERD measurements match on the probed snow blocks, did they use enough CT samples to provide a reasonable estimate of the spatial variability within the snow volume probed by the NERD, did they only use the very top of the snow samples for their CT analysis because only this part is probed by the NERD due to the long NIR wavelengths of its illumination sources?

2)

Without validation of NERD SSA measurements, and thus without an essential part of any presentation of a novel measurement method, some of the made statements seem rather unfounded. For example, the authors seem to repeatedly suggest that the NERD can accurately measure BRFs for snow, but how do they know that? Only BRF measurements on reflectance standards are somewhat validated as their nominal reflectance values are known and as they are roughly compared in this manuscript to reflectance measurements of reflectance standards that were performed with previously presented measurement methods. Going on to claim accurate snow SSA measurements based on these basic results and without any quantitative validation seems to be even more unfounded than claiming accurate snow BRF measurements.

The authors also stress the high cost of other snow SSA measurement methods and the low cost of their anticipated NERD instrument. Yet, I fail to see how the development of a useful measurement tool in the field based on their measurement principle will lead to a price of the instrument of less than thousands of USD, similar to the cost of some of the other optical measurement methods that can be used to derive snow SSA. Can the authors give a realistic cost estimate to justify such claims, factoring in development, prototyping, weather sealing, installation of permanent and sturdy components, … of a portable NERD measurement tool? If they mean the NERD will be cheaper than a CT or a high-resolution spectrometer, then they should not overly generalize by including all previously presented measurement methods in this statement, while they actually only talk about some of those (see also point 3 below).

3)

Particularly bothersome are some generalizations and omissions when discussing previously presented measurement methods and the motivation behind and the potential benefits of the NERD. Fast and nondestructive snow SSA measurements can be obtained with the InfraSnow (introduced by Gergely et al. 2014) or by contact spectroscopy (introduced by Painter et al. 2007), for example. No sampling is required, no samples are destroyed. In fact, the InfraSnow was developed for some of the same reasons and applications as the NERD, as stated by Gergely et al. (2014), and its presentation additionally included a quantitative uncertainty analysis and measurement validation, yet none of this is mentioned in this manuscript. Instead, all previously presented measurement methods for deriving snow SSA are falsely lumped together, and the manuscript gives the impression that the anticipated NERD method will be the first and only fast, nondestructive snow SSA measurement method, for example. This is poor 'scientific' work.

The authors should either summarize and discuss the various measurement methods separately and in much more detail without overly generalizing and thus without making misleading and false claims, or they should simply state that they are attempting to develop a novel measurement tool that allows fast, nondestructive SSA measurements without suggesting that such measurements are

not possible with (any of the) previously developed measurement methods. Also, NIR photography should be included as a reference in the list of currently available optical SSA measurement methods:

```
@ARTICLE{Matzl+06,
  author = {M. Matzl and M. Schneebeli},
  title = {Measuring specific surface area of snow by near-infrared photography},
  journal = jg,
  year = {2006},
  volume = {52},
  pages = {558--564},
  number = {179},
  doi = {10.3189/172756506781828412}
}
```

4)
The main results in the context of the presentation of the NERD measurement method are summarized in Fig. 6. The effect of relevant sources of uncertainty should be discussed quantitatively instead of only stating the variability of the fitted exponential function for the extremely limited number of samples used to derive the fit (see point 1).

Here, another effect is of interest: The small penetration depth of just a few mm or less at long NIR wavelengths is clearly much shallower than the penetration depth of visible light which forms the main contribution to overall solar irradiation. So, if snow surface SSA is measured at long NIR wavelengths, how realistic is it to analyze overall snow albedo based on the derived SSA value?

It may be that the top few mm of the snow cover that determine long-wavelength NIR reflectance do not represent the full near-surface snow that determines overall snow albedo, e.g., a thin layer of surface hoar or some very small windblown snow fragments deposited at the very surface. Such a discussion would also add further scientific value to the study beyond presenting a novel SSA measurement method.

5)
Terminology: Throughout the manuscript, 'error' should be changed to 'uncertainty' or 'difference', depending on the context. Usage of 'error' when actually talking about 'uncertainty' or simple 'differences' is deprecated in measurement science and avoided to guarantee a more precise and meaningful terminology (see the 'Guide to the expression of uncertainty in measurement'), especially as a realistic assessment of uncertainties (and not errors) becomes increasingly important in remote sensing and climate modeling applications.

Specific comments:

page 1 line 1: Is SSA an important physical property because it directly affects solar radiation, or is it important for another reason and it also affects solar radiation? The authors should specify this.

page 1 line 2f: This is a misleading generalization, if no further details are given about the various different SSA measurement methods (see point 3 above). The authors should remove this sentence and instead focus on the analysis and description of the NERD measurement principle in the abstract, and, e.g., on its possible future use to track snow SSA evolution on a time scale of hours.

page 1 line 17f: I do not understand this sentence. What is 'positive snow internal albedo feedback'? Is this the same as 'positive albedo feedback' in the previous sentence? This sentence could probably be rewritten to clarify.

p.2 l.4: What 'particles'? Snow is not a granular material of individual particles or grains but a material characterized by a complex 3D microstructure with continuous ice and air phases. Probably this should be the 'snow microstructure' or are the authors talking specifically about modeling snow here as a matrix of suspended ice particles?

p.2. l.22: What is 'grain growth'? Snow is not a granular material. 'grain growth' could simply be left out. … where solar heating induces a further decrease in SSA, …

p. 3 l. 1 – 6: This is a misleading generalization (see point 3 above). Discuss different methods separately and in more detail, or simply state (1) that different methods to measure snow SSA for different applications have been presented previously (including the corresponding references) and (2) that this study describes a novel measurement method for fast and nondestructive SSA measurements (without trying to motivate it by making misleading and false claims about other measurement methods). Readers can then go back to the cited studies for details and see for themselves how different measurement methods compare to each other and what may be an advantage or disadvantage for different applications.
Because a short and still adequate, i.e., not misleading or false, description of all relevant previously presented SSA measurement methods may be difficult to achieve within a few of lines of text, the following approach could be used:
keep lines 1 – 3 and add reference Matzl and Schneebeli 2006 (see point 3 above), delete lines 4 – 6, keep line 6f and add: … is widely sought after, which not all (or which only few) previously presented measurement methods allow. Here, we introduce ...

p. 4 l. 18 – 25: This is not a validation for determining BRFs of snow with the NERD due to the very different nature of snow (uneven surface, extensive volume-scattering). So, l. 24f would be more correctly rephrased as: … to obtain BRFs on smooth reflectance standards with a measurement uncertainty of …, or simply delete this statement.

p. 4 l. 32: pixel (2D) or voxel (3D equivalent of pixel)?

p.4 l. 33f: Can the authors give the temperature of the CT and CT sample more accurately than below 0°C, if snow can't survive much more than 15 min? How does this affect snow SSA evolution during the duration of CT measurements? Has this CT been used for snow SSA measurements previously, is the CT resolution high enough to yield reliable snow SSA values, have previous CT measurements of snow with this CT been validated against other measurement methods (or other CT setups)? This information should be included in the text.

p. 5 l. 10: Delete 'relatively'.

p. 5 l. 20: What is 'highly sintered' snow. Old hard snow?

p. 6 l. 9: What is 'visibly apparent snow metamorphosis'? Is this temperature-gradient metamorphism or equal-temperature metamorphism or both, resulting in what type of snow (e.g., depth hoar or melt-freeze or other)? The authors should specify this in the text or simply list the physical properties of the snow sample and refer the reader to the CT images for further information.

p. 6 l. 13ff: Has this Monte Carlo model been validated or at least used for snow previously? Is there any indication of what the expected uncertainty is for applying the model to snow (and not only to Lambertian surface scattering as described on p.7)? Such information should be included in the text, if it is available.

p. 6 l. 23: Why are at least 100 thousand photons used per simulation, while commonly millions of photons are needed for complex Monte Carlo raytracing simulations. Have the authors checked that an increase in photons beyond the photon numbers that they have chosen does not lead to significant changes in the Monte Carlo modeling results for snow? If this is the case both for the tested Lambertian surfaces and for Monte Carlo simulations for volume-scattering snow, they should state this in the text. Or they should state how much a further increase in the number of photons may change the modeling results for snow.

p. 6 l. 24ff: Are the ice particle scattering properties obtained for randomly oriented or preferentially oriented ice particles (horizontally, vertically, something else?) within the snow matrix? How realistic is this assumption for the analyzed snow types? I would intuitively expect that random orientation should be the most realistic assumption in general, but some snow does show strong anisotropy. This information should be included in the manuscript.
Similarly, are all ice particle types equally realistic representations of natural snow? Particularly hexagonal plates seem rather extreme when intuitively compared to the 3D microstructure of natural snow, which is also confirmed in Figs. 4 and 5. Maybe it would be more realistic not to include hexagonal plates in the analysis, which could also streamline the discussion?

p. 6 l. 29f: Delete this sentence, it is a duplicate of l. 22f.

p. 7 l. 14ff: How are the stark differences in CT and NERD measurement volumes included in the analysis? How does this affect the analysis results (see also point 1 above)? This should be discussed in the text.

p. 7 l. 24: How can this be claimed without validating snow BRF measurements with the NERD against any other independent snow BRF measurements, e.g., a gonioreflectometer, and without providing any quantification of the term 'relatively accurate' when comparing NERD BRF measurements and modeled BRFs? The authors should either delete this sentence or provide a detailed quantification instead of a vague statement.

p. 8 l. 10f: Better: With the InfraSnow, Gergely et al. (2014) were able to determine the reflectance values of nominal 25 %, 50 %, and 99 % reflectance standards to within an accuracy of better than 1 %.

p.8 l. 11: 'directional-hemispherical reflectance' is not correct here due to the diffusing cone in the InfraSnow that prevents direct illumination of the snow surface and instead guarantees predominantly diffuse illumination. 'directional-' should be removed.

p. 8 l. 12 − 16: Best to remove these two sentences. Lambertian reflectance standards are only part of the testing performed for the InfraSnow. Additionally, various other surfaces, including snow, are tested. This should be included if l.12,13 are kept in the text. The second sentence does not add important information and is highly speculative, especially when trying to translate the results found for reflectance standards to reflectance measurements on snow, due to the very different nature of NIR light scattering in snow and the differences in the applied measurement techniques (see also point 1 above).

p. 8 l. 28ff: How does this BRF uncertainty affect the derived snow SSA?
And how could light leakage to and from the outside of the NERD due to an uneven snow surface or due to specular reflections also add to the uncertainty in BRF measurements (see also point 1 above)?

p. 8 l. 32ff: The authors should indicate which Figure or Table they are referring to.

p.9 l. 1f: What is an estimate of this uncertainty then? Can the authors provide a quantification? Otherwise, it is better not to include such statements.

p.10 l.3 – 18: I do not see the immediate relevance of this discussion. This could be mostly removed to streamline the manuscript and focus on more crucial results, which are given in the next paragraph. I would prefer to see the authors try to include an actual quantitative uncertainty assessment of their measurement method, including the effect of grain shape on their modeling results instead of this ancillary discussion.

p. 10 l. 22: I do not see a compelling reason in the presented data why this relationship has to be an exponential function. Could it be something else (linear relationship), and would this significantly change any of the results? If so, this effect should be included in the SSA measurement uncertainty. Also, what does a constant SSA measurement uncertainty across the entire SSA range mean for using the NERD to measure different snow types? High relative SSA measurement uncertainty for snow characterized by low SSA (what snow types are those?) and low relative SSA measurement uncertainty for snow characterized by high SSA (what snow types are those?). This should be added to the text to illustrate the uncertainty in the derived BRF-to-SSA relationship beyond the mere presentation of the values for the root mean square differences.

p. 11 l. 9: Delete 'accurate'. There is no quantitative validation of the SSA measurements in the manuscript. 'quick, reliable, and repeatable' convey the full picture. Even 'repeatable' could be removed because it is implied by 'reliable'.

p. 11 l. 16,17,18: The authors should replace 'will' with 'can' or 'could' or 'may', or preface each of these statements by 'We believe that ….' or 'We intend to use the NERD for ...' or 'The analysis indicates the potential for ….' or similar. Selling such statements as foregone conclusions seems like a far stretch given the limited analysis in the manuscript and the inherent uncertainty of future developments (see also point 1 above).

Caption Figure 3: Snow is not a granular material. Better: … 'as the snow microstructure gets coarser' or 'is characterized by more rounded shapes'.

---

## Author Comment (AC1) · 5 Mar 2019

March 4, 2019

Dear Dr. Chambon,

We are grateful for the referees' time spent providing helpful comments and suggestions. In response to their reviews, we restructured and changed the title of the manuscript (tc-2018-198) to *Monitoring of Snow Surface Near-Infrared Bidirectional Reflectance Factors with Added Light Absorbing Impurities*. Furthermore, we now present our key scientific results that were originally going to appear in a separate paper.

In the following attachment, we respond to the referees' comments. We begin with our response to Anonymous Referee #3's general and specific comments. Next we address the detailed comments by Anonymous Referee #2. Finally, we respond to specific and minor comments from Dr. Dumont. We hope you find that our revised manuscript addresses the main concerns, inherent problems, and recommendations raised by all three referees, some of which are no longer relevant.

We look forward to your final decision regarding the acceptance of this manuscript for publication in *The Cryosphere*.

Sincerely,

Adam Schneider
amschne@umich.edu

We are thankful for Anonymous Referee #3's review. To summarize, Referee #3 brought to our attention three critical flaws with the manuscript. First, they identify that the overall tone of the manuscript presents the The Near-Infrared Emitting and Reflectance Monitoring Dome (NERD) as a novel instrument that will eliminate the need for other similar instruments. They also point out an alarming over-simplification in the background presentation of previous snow specific surface area (SSA) measurement methods. These critical flaws are, to some extent, inherent from a lack of quantitative uncertainty analysis in the results and discussion. We generally agree with these criticisms and rewrote the manuscript accordingly.

In addition to renaming the manuscript, we refocused the primary objective on better understanding how light absorbing impurities (LAIs) affect snow albedo feedbacks. The revised manuscript better presents the NERD as an instrument that measures snow bidirectional reflectance factors to approximate snow SSA. Here, we apply the NERD specifically for the purpose of monitoring hourly scale snow surface microphysical properties with and without added LAIs. In light of this repurposing, the revised manuscript avoids language that implies that the NERD is validated as a precise snow SSA measurement method. Furthermore, we added new results from our LAI in snow experiments and discuss our findings in the context of the NERD's limitations. Following are our responses to Reviewer #3's comments, which are italicized for reference:

1. *Validation of the NERD measurement principle would entail comparing snow SSA values measured with the NERD to independent SSA measurements, e.g., SSA measured by other optical measurement methods or by methane gas absorption (or at the very least a vast number of CT measurements that are not used for developing the NERD measurement method). The authors have merely shown the consistency of BRF measurements (not SSA measurements) performed with the NERD on highly uniform, diffusely scattering reflectance standard surfaces. Neither the effects of extensive volume-scattering in snow nor possible directional/specular effects at the snow surface are included here. For example, is there light leakage into or out of the NERD on uneven surfaces, and what effect do oriented surface structures (small dents or ripples on the surface) have on measured BRF (I would expect this to be of greater concern here than for previously presented methods that are based on diffuse instead of directional reflectance)?*
*Furthermore, no systematic and quantitative uncertainty assessment of snow SSA measurements with the NERD is presented (although the impact of various sources of uncertainty on measured BRF is discussed), which is an integral part of any presentation of a novel measurement method, as described in the 'Guide to the expression of uncertainty in measurement', for example. Particularly an inter-comparison of multiple measurement methods or comparisons of in situ and remote sensing observations, as alluded to in the conclusions of the manuscript, require a thorough assessment of the uncertainties affecting each measurement method to allow drawing reliable conclusions from such comparisons. In this manuscript, only a rough estimate of the variability of an empirically derived exponential relationship between BRF and SSA is interpreted as SSA measurement uncertainty, based on a low number of sample measurements. For empirical measurement methods, i.e., measurement methods that are not based on physical measurement models but on statistical correlation as presented in this manuscript, a large number of samples is required to guarantee a high statistical significance. A low number of samples can easily lead to highly underestimated (or overestimated)*

*measurement uncertainties over the entire or parts of the measurement range, especially when not validated against independent measurement methods.*

*A detailed NERD SSA measurement uncertainty assessment and validation would also include how the mismatch between CT measurement volume and NERD measurement volume affects the analysis. While suggesting to use the NERD to determine the large-scale spatial variability of snow SSA, the authors fail to discuss this very effect at a small scale of 10 cm, relevant for the NERD measurement principle, e.g., how do CT samples and NERD measurements match on the probed snow blocks, did they use enough CT samples to provide a reasonable estimate of the spatial variability within the snow volume probed by the NERD, did they only use the very top of the snow samples for their CT analysis because only this part is probed by the NERD due to the long NIR wavelengths of its illumination sources?*

First, the lack of an independent SSA measurement method to validate NERD SSA measurements is a fundamental flaw in the presentation of the instrument. In the revised manuscript, we add limited results from contact spectroscopy measurements to determine optically derived snow SSA. These new results are included in Fig. 5. and discussed in the text toward the end of section 3.1.

Second, the effects of extensive volume-scattering within the snow are ignored in the context of the NERD snow BRF retrieval. Neglecting extensive volume scattering in snow, while it may be more relevant to other wavelengths, is purposeful here. Volume scattering is explored using three dimensional Monte Carlo modeling. In fact, these modeling results indicate that photons at 1.30 μm (and 1.55 μm) undergo an order of magnitude or two fewer scattering events than shorter, visible wavelengths. Furthermore, we estimate that the most of these photons' path lengths are limited to just a couple centi-meters within the upper-most layers of the snow pack, as demonstrated by the below histograms.

[Figure]

These histograms show Monte Carlo photon path lengths simulated in snow. In the model, snowpack is represented by a matrix of homogenous, randomly oriented aspherical ice particles suspended in air. Particle sizes (100, 300 μm) are defined by their sphere equivalent radii, calculated from their projected areas.

Because of these results, we assume that in the most general case, the snow bidirectional

scattering-surface reflectance-distribution function (which includes subsurface volume scattering) is well approximated by the simpler snow bidirectional reflectance distribution function. While this assumption would not be valid for visible and shorter near-infrared wavelengths, we believe this assumption is valid at 1.30 and 1.55 µm. This is also why we often use the term "reflectance factor" to describe our measurements.

Excess ambient Light into the field of view of the NERD photodiodes is accounted for by subtracting dark current from photodiode current measurements. Measurements are collected continuously so that the displayed BRF is representative of the previous 5-10 seconds worth of measurements. Therefore, this procedure yields accurate BRF measurements only after not moving for roughly 5 seconds. If the environmental conditions are changing rapidly, e.g., when clouds are moving in and out of view of the sun, then BRF measurements are unreliable.

Micro-scale ripple / lens effects strong enough to have a measurable impact on the NERD are of interest but beyond the scope of this study. Assuming these effects lead to measurement uncertainty for directional reflectance retrieval, we attempt to minimize this uncertainty by using multiple infrared emitters and photodiodes spanning four zenith / azimuth angle combinations. In the manuscript, all snow BRFs presented represent medians or means of as many as eight samples obtained from two independent viewing azimuth angles and also from rotating the dome.

Third, the manuscript is poorly structured and is unfocused. This ambiguous purpose leads readers to believe that the results and discussion present the NERD as an instrument capable of obtaining precise snow SSA in applications beyond the scope of the revised manuscript. While accurate, precise snow BRFs can be measured in favorable conditions, limited evaluation and validation of snow SSA retrieval cannot be assumed from the presented results. In response to this critical flaw, we repurposed the entire study, as stated previously, and limit our discussion of NERD derived snow SSA results to our LAI in snow experiments. We present these results in the context of a large uncertainty range, provided by the error bars in Figs. six and seven, and alongside CT results for comparison. As pointed out by the referee, a full NERD snow SSA validation would entail far more samples and additional measurement methods.

Because the primary study of the revised manuscript is on LAIs' influence on snow albedo feedbacks, we only partially address the above criticism by showing a small number of data points from contact spectroscopy measurements in Figure 5. These SSA data are colored consistently with the snow samples they represent, but marked by hollow triangles for depth hoar (blue) and rounded grains (pink).

Finally, the manuscript fails to address how closely the NERD-probed snow samples relate to those collected and placed into the micro-CT machine. This point highlights one of the main difficulties associated with the comparison of NERD snow measurements to micro-CT snow samples. Getting the same snow that was targeted with NERD into the micro-CT machine is challenging, yielding inherent uncertainty in the precise determination of snow SSA using the presented NERD calibration function. Without access to other instrumentation that will operate in the field, it is nearly impossible to perform an apples to apples comparison of snow measurements. This is also one of the main motivations for developing this instrument. But a more complete SSA calibration remains challenging.

In response to these inherent uncertainties, we attempted to collect just the top few centimeters of snow closest to that probed by the NERD, as suggested by the referee. For best results, snow samples collected just outside of the Cold Regions Research Engineering Laboratory were transported directly into the micro-CT machine immediately following NERD measurements. Ironically, results from these "best" comparisons yield weaker correlations between NERD BRFs and CT SSA than our other comparisons.

Because of the inherent uncertainty with deriving snow SSA from NERD BRF measurements using X-CT for calibration, the revised manuscript generally avoids discussion pertaining to the precise determination and demonstration of snow SSA. Instead, we now focus on the demonstration of large, highly likely, changes in surface snow behavior at large after adding LAIs. These significant changes are detectable by the NERD and are complemented by CT.

2. *Without validation of NERD SSA measurements, and thus without an essential part of any presentation of a novel measurement method, some of the made statements seem rather unfounded. For example, the authors seem to repeatedly suggest that the NERD can accurately measure BRFs for snow, but how do they know that? Only BRF measurements on reflectance standards are somewhat validated as their nominal reflectance values are known and as they are roughly compared in this manuscript to reflectance measurements of reflectance standards that were performed with previously presented measurement methods. Going on to claim accurate snow SSA measurements based on these basic results and without any quantitative validation seems to be even more unfounded than claiming accurate snow BRF measurements.*

*The authors also stress the high cost of other snow SSA measurement methods and the low cost of their anticipated NERD instrument. Yet, I fail to see how the development of a useful measurement tool in the field based on their measurement principle will lead to a price of the instrument of less than thousands of USD, similar to the cost of some of the other optical measurement methods that can be used to derive snow SSA. Can the authors give a realistic cost estimate to justify such claims, factoring in development, prototyping, weather sealing, installation of permanent and sturdy components, ... of a portable NERD measurement tool? If they mean the NERD will be cheaper than a CT or a high-resolution spectrometer, then they should not overly generalize by including all previously presented measurement methods in this statement, while they actually only talk about some of those (see also point 3 below).*

First, we believe the NERD is capable of obtaining accurate snow BRFs at 1.30 and 1.55 μm because of the relatively small photon path lengths here compared to shorter wavelengths. This reasoning is fully addressed in the response to comment one. In short, assuming that the snow bidirectional surface-scattering reflectance-distribution function is well approximated by the snow bidirectional reflectance distribution function, the effects of volume scattering can be ignored. In response to this point, we renamed the revised manuscript to emphasize surface snow bidirectional reflectance factor measurements. Reflectance factors are useful here, because by definition, they are comparison measurements to ideal Lambertian reflectors.

Second, the manuscript overgeneralizes in making reference to previous snow SSA measurement methods. As a result, readers are to assume that the anticipated NERD instrument will be better and cheaper than previous methods. For this reason, we rewrote the entire manuscript to focus on our scientific results. We intend to present the NERD measurement method only in the context of monitoring snow surface BRFs to study how LAI affect snow

albedo feedbacks and snow metamorphism.

We cannot, at this time, give a full cost estimate for the production quality version of the NERD. We were able to create two functioning prototypes, which are not fully weather-proofed, insulated, or durable (although it has endured multiple flights, car trips, and field campaigns) for roughly 500USD.

3. *Particularly bothersome are some generalizations and omissions when discussing previously presented measurement methods and the motivation behind and the potential benefits of the NERD. Fast and nondestructive snow SSA measurements can be obtained with the InfraS-now (introduced by Gergely et al. 2014) or by contact spectroscopy (introduced by Painter et al. 2007), for example. No sampling is required, no samples are destroyed. In fact, the InfraSnow was developed for some of the same reasons and applications as the NERD, as stated by Gergely et al. (2014), and its presentation additionally included a quantita-tive uncertainty analysis and measurement validation, yet none of this is mentioned in this manuscript. Instead, all previously presented measurement methods for deriving snow SSA are falsely lumped together, and the manuscript gives the impression that the anticipated NERD method will be the first and only fast, nondestructive snow SSA measurement method, for example. This is poor 'scientific' work. The authors should either summarize and discuss the various measurement methods separately and in much more detail without overly gen-eralizing and thus without making misleading and false claims, or they should simply state that they are attempting to develop a novel measurement tool that allows fast, nondestructive SSA measurements without suggesting that such measurements are not possible with (any of the) previously developed measurement methods. Also, NIR photography should be included as a reference in the list of currently available optical SSA measurement methods:*
*@ARTICLE{Matzl+06,*
*author = {M. Matzl and M. Schneebeli},*
*title = {Measuring specific surface area of snow by near-infrared photography},*
*journal = jg,*
*year = {2006},*
*volume = {52},*
*pages = {558–564},*
*number = {179},*
*doi = {10.3189/172756506781828412}*

We added the missing references to the introduction. Furthermore, we provided more details regarding each instrument technique. In the revised manuscript, we keep these references concise to instead focus on the specific background information pertinent to the new purpose (i.e., snow metamorphism in the presence of LAIs).

4. *The main results in the context of the presentation of the NERD measurement method are summarized in Fig. 6. The effect of relevant sources of uncertainty should be discussed quantitatively instead of only stating the variability of the fitted exponential function for the extremely limited number of samples used to derive the fit (see point 1). Here, another effect is of interest: The small penetration depth of just a few mm or less at long NIR wavelengths is clearly much shallower than the penetration depth of visible light which forms the main contribution to overall solar irradiation. So, if snow surface SSA is measured at long NIR*

*wavelengths, how realistic is it to analyze overall snow albedo based on the derived SSA value? It may be that the top few mm of the snow cover that determine long-wavelength NIR reflectance do not represent the full near-surface snow that determines overall snow albedo, e.g., a thin layer of surface hoar or some very small windblown snow fragments deposited at the very surface. Such a discussion would also add further scientific value to the study beyond presenting a novel SSA measurement method.*

Thank you for bringing this interesting point regarding snow albedo to our attention. As mentioned, because the NERD only probes a thin surface layer of the snow, these measurements are not the best representation of snow albedo at large. Therefore, we removed language that implies that snow infrared BRFs are a good proxy for snow broadband albedo.

5. *Terminology: Throughout the manuscript, 'error' should be changed to 'uncertainty' or 'difference', depending on the context. Usage of 'error' when actually talking about 'uncertainty' or simple 'differences' is deprecated in measurement science and avoided to guarantee a more precise and meaningful terminology (see the 'Guide to the expression of uncertainty in measurement'), especially as a realistic assessment of uncertainties (and not errors) becomes increasingly important in remote sensing and climate modeling applications.*

As recommended, we changed "error" to "uncertainty" or "difference" where appropriate throughout the revised manuscript.

6. *page 1 line 1: Is SSA an important physical property because it directly affects solar radiation, or is it important for another reason and it also affects solar radiation? The authors should specify this.*

We removed this confusing sentence from the abstract.

7. *page 1 line 2f: This is a misleading generalization, if no further details are given about the various different SSA measurement methods (see point 3 above). The authors should remove this sentence and instead focus on the analysis and description of the NERD measurement principle in the abstract, and, e.g., on its possible future use to track snow SSA evolution on a time scale of hours.*

We removed the misleading sentence and over-generalization regarding other snow SSA measurement methods. In the revised abstract, we focus on the NERD as a tool to monitor surface BRFs of snow with and without large LAI concentrations.

8. *page 1 line 17f: I do not understand this sentence. What is 'positive snow internal albedo feedback'? Is this the same as 'positive albedo feedback' in the previous sentence? This sentence could probably be rewritten to clarify.*

We repurposed the introduction section. As a result, mention of snow internal albedo feedback is delayed until more specific background presented later in the introduction.

9. *p.2 l.4: What 'particles'? Snow is not a granular material of individual particles or grains but a material characterized by a complex 3D microstructure with continuous ice and air phases. Probably this should be the 'snow microstructure' or are the authors talking specifically about modeling snow here as a matrix of suspended ice particles?*

Thank you for pointing out a confusing part of our definition of snow SSA. We changed the wording around eq. 1 to apply to snow as a porous ice / air microstructure, instead of what it represents in particle based snow models.

10. *p.2. l.22: What is 'grain growth'? Snow is not a granular material. 'grain growth' could simply be left out. ... where solar heating induces a further decrease in SSA, ...*

We removed this sentence.

11. *p. 3 l. 1 – 6: This is a misleading generalization (see point 3 above). Discuss different methods separately and in more detail, or simply state (1) that different methods to measure snow SSA for different applications have been presented previously (including the corresponding references) and (2) that this study describes a novel measurement method for fast and non-destructive SSA measurements (without trying to motivate it by making misleading and false claims about other measurement methods). Readers can then go back to the cited studies for details and see for themselves how different measurement methods compare to each other and what may be an advantage or disadvantage for different applications. Because a short and still adequate, i.e., not misleading or false, description of all relevant previously presented SSA measurement methods may be difficult to achieve within a few of lines of text, the following approach could be used: keep lines 1 – 3 and add reference Matzl and Schneebeli 2006 (see point 3 above), delete lines 4 – 6, keep line 6f and add: ... is widely sought after, which not all (or which only few) previously presented measurement methods allow. Here, we introduce ...*

We removed the misleading generalizations and expanded on the discussion of the most relevant techniques. The restructured manuscript at large also relieves these previously fundamental flaws with the previous introduction.

12. *p. 4 l. 18 – 25: This is not a validation for determining BRFs of snow with the NERD due to the very different nature of snow (uneven surface, extensive volume-scattering). So, l. 24f would be more correctly rephrased as: ... to obtain BRFs on smooth reflectance standards with a measurement uncertainty of ..., or simply delete this statement.*

Following this suggestion, we changed this sentence to include "...obtain BRFs on smooth reflectance standards.... (sec. 2.1, par. 4)"

13. *p. 4 l. 32: pixel (2D) or voxel (3D equivalent of pixel)?*

Voxels for 3D reconstructions, pixel for 2D cross sections. We clarified this in the revised manuscript.

14. *p.4 l. 33f: Can the authors give the temperature of the CT and CT sample more accurately than below 0°C, if snow can't survive much more than 15 min? How does this affect snow SSA evolution during the duration of CT measurements? Has this CT been used for snow SSA measurements previously, is the CT resolution high enough to yield reliable snow SSA values, have previous CT measurements of snow with this CT been validated against other measurement methods (or other CT setups)? This information should be included in the text.*

CT scans were conducted in a cold lab at roughly 27 degrees Fahrenheit. Snow SSA evolution over the course of 15 mins is assumed to be minimal, which is why we report the

technical specifications of the CT in the text. Yes, the CT machine has been used for SSA measurements previously. We added a reference to Lieb-Lappen et al. (2017), who provide a thorough presentation of the CT methodology for ice samples. Here, we applied their methods to snow samples and calculated snow SSA according to Pizner and Schneebeli (2009), whom we also cite in the text. Based on the volume rendering images, which clearly show the finer scale features of the higher SSA needles, we believe that the resolution is high enough to derive reliable SSA, although we are unsure of the range of uncertainty that these algorithms yield. While the snow images in (previous) Fig. 3 can facilitate this assumption, we removed the images from the revised manuscript to streamline the main messages.

15. *p. 5 l. 10: Delete 'relatively'.*

We folded, condensed, and rewrote the snow samples descriptions section. In the revised manuscript, we reclassify the samples according to Fierz et al. (2009). We also removed the word "relatively."

16. *p. 5 l. 20: What is 'highly sintered' snow. Old hard snow?*

We now describe snow physical parameters according to Fierz et al. (2009). As a result, we removed this description.

17. *p. 6 l. 9: What is 'visibly apparent snow metamorphosis'? Is this temperature-gradient metamorphism or equal-temperature metamorphism or both, resulting in what type of snow (e.g., depth hoar or melt-freeze or other)? The authors should specify this in the text or simply list the physical properties of the snow sample and refer the reader to the CT images for further information.*

Physical properties are now provided in Table 2 according to Fierz et al. (2009). We removed this confusing description.

18. *p. 6 l. 13ff: Has this Monte Carlo model been validated or at least used for snow previously? Is there any indication of what the expected uncertainty is for applying the model to snow (and not only to Lambertian surface scattering as described on p.7)? Such information should be included in the text, if it is available.*

This Monte Carlo model is applied in a few previous studies to study light penetration in snow (e.g. Smith et al. (2018)). The model best approximates a very dense ice cloud with small suspended aspherical ice particles. Therefore, it is difficult to quantitatively estimate the uncertainty associated with its results applied to snow. Here, we apply the model to study directional reflectance. For validation, we provide comparisons with the SNICAR model for spheres. Albedo calculated for spheres agrees with that calculated with the SNICAR model. This comparison is provided in the text and data are shown in Fig. 3.

19. *p. 6 l. 23: Why are at least 100 thousand photons used per simulation, while commonly millions of photons are needed for complex Monte Carlo raytracing simulations. Have the authors checked that an increase in photons beyond the photon numbers that they have chosen does not lead to significant changes in the Monte Carlo modeling results for snow? If this is the case both for the tested Lambertian surfaces and for Monte Carlo simulations for volume-scattering snow, they should state this in the text. Or they should state how much a further increase in the number of photons may change the modeling results for snow.*

We did not clarify this previously in the text. In the revised manuscript we touch on this, but please see the below modeling results here for further information.

[Figure]

In the left column, we show BRFs calculated for snow from a simulation of 100,000 photons at 3x3 degree resolution. Azimuthally dependent BRFs are too noisy for meaningful interpretations. Azimuthal averaging (bottom row), reduces Monte Carlo noise, but not sufficiently for useful comparisons to NERD measurements.

In the right column, we show BRFs calculated for snow from a simulation of 10,000,000 photons (also at 3x3 degree resolution). Here, the specular reflection feature can be faintly seen (for an illumination zenith angle of 20 degrees). This is a good indication that Monte Carlo noise is sufficiently small. As expected, azimuthal averaging (bottom row) removes almost all Monte Carlo noise.

In the revised manuscript, all Monte Carlo BRFs presented are calculated from simulations of 1,000,000 photons and are azimuthally averaged. While azimuthally dependent results are great for data visualization, they do not provide any additional information, as azimuthal directional scattering is determined at random from a uniform probability density function ranging from 0 to $2\pi$. Therefore, BRFs are azimuthally symmetric.

In further testing, we find that it is best use 10,000,000 photons to generate the figures shown above. With azimuthal averaging, however, BRFs stabilize for simulations of 250,000 to 500,000 photons. Therefore, in the revised manuscript, we show only results presented for simulations of 1,000,000 photons, which are sufficiently stable.

20. *p. 6 l. 24ff: Are the ice particle scattering properties obtained for randomly oriented or preferentially oriented ice particles (horizontally, vertically, something else?) within the*

*snow matrix? How realistic is this assumption for the analyzed snow types? I would intuitively expect that random orientation should be the most realistic assumption in general, but some snow does show strong anisotropy. This information should be included in the manuscript. Similarly, are all ice particle types equally realistic representations of natural snow? Particularly hexagonal plates seem rather extreme when intuitively compared to the 3D microstructure of natural snow, which is also confirmed in Figs. 4 and 5. Maybe it would be more realistic not to include hexagonal plates in the analysis, which could also streamline the discussion?*

The single scattering properties are obtained for randomly oriented ice particles. This is now clarified in the text. As mentioned previously, the model best represents a very dense ice cloud. It is not necessarily the best representation for snow, but its purposes are to (a) approximate the relationships between snow SSA and BRFs at 30 and 60 degrees viewing and (b) to explore how much variability we might expect for different snow types. While previous snow albedo models use idealized spherical particle surfaces, here, we are interested in exploring how BRFs change when we apply full scattering phase functions from aspherical ice particles.
The hexagonal plates yield consistently lower BRFs due to their (even more) extreme forward scattering peaks. In response to this comment, we removed the plates from the results and discussion.

21. *p. 6 l. 29f: Delete this sentence, it is a duplicate of l. 22f.*

Thank you for pointing out this editing oversight. We removed both of these sentences in the revised manuscript.

22. *p. 7 l. 14ff: How are the stark differences in CT and NERD measurement volumes included in the analysis? How does this affect the analysis results (see also point 1 above)? This should be discussed in the text.*

This comment highlights one of our main concerns with our calibration approach, and could potentially be one of the main sources of uncertainty regarding the precise determination of snow SSA using the NERD. We touched on this point in our response to point 1 above. In short, we attempt to mitigate these uncertainties by sampling just the uppermost layers of the snow pack. To highlight this approach we changed the title of the manuscript to include "surface."

23. *p. 7 l. 24: How can this be claimed without validating snow BRF measurements with the NERD against any other independent snow BRF measurements, e.g., a gonioreflectometer, and without providing any quantification of the term 'relatively accurate' when comparing NERD BRF measurements and modeled BRFs? The authors should either delete this sentence or provide a detailed quantification instead of a vague statement.*

We rewrote the results and discussion section. Therefore, this comment is only generally relevant, but still important. In theory, a gonioreflectometer would be subject to the same sources of bidirectional reflectance measurement uncertainty in the case of extensive volume scattering. Because we expect extensive volume scattering to be minimal at wavelengths of interest (see also response to comment #1), we believe the NERD gives accurate BRF

measurements, which are directly compared to Lambertian reflectance targets in frequent calibration.

24. *p. 8 l. 10f: Better: With the InfraSnow, Gergely et al. (2014) were able to determine the reflectance values of nominal 25 %, 50 %, and 99 % reflectance standards to within an accuracy of better than 1 %.*

Due to our repurposing, this comparison is less relevant to the revised manuscript results and discussion. Therefore, we removed it.

25. *p.8 l. 11: 'directional-hemispherical reflectance' is not correct here due to the diffusing cone in the InfraSnow that prevents direct illumination of the snow surface and instead guarantees predominantly diffuse illumination. 'directional-' should be removed.*

Thank you for pointing out the incorrect description of reflectance measurements conducted by the Infrasnow. We corrected this description in the introduction and removed this discussion from this section.

26. *p. 8 l. 12 – 16: Best to remove these two sentences. Lambertian reflectance standards are only part of the testing performed for the InfraSnow. Additionally, various other surfaces, including snow, are tested. This should be included if l.12,13 are kept in the text. The second sentence does not add important information and is highly speculative, especially when trying to translate the results found for reflectance standards to reflectance measurements on snow, due to the very different nature of NIR light scattering in snow and the differences in the applied measurement techniques (see also point 1 above).*

We removed these sentences.

27. *p. 8 l. 28ff: How does this BRF uncertainty affect the derived snow SSA? And how could light leakage to and from the outside of the NERD due to an uneven snow surface or due to specular reflections also add to the uncertainty in BRF measurements (see also point 1 above)?*

BRF uncertainties will propagate through to SSA calculations. These uncertainties are included in the error bars in NERD SSA results in figs. 6 and 7. Light leakage into the dome saturates the photodiode sensors, making measurements in diffuse lighting conditions difficult this point is discussed further in the text.
Also, because dark currents are subtracted from photodiode currents every measurement cycle, static ambient light leakage into the dome corrected for in BRF calculations. In fast-changing ambient lighting conditions, measurements are not reliable (see also response to point 1 above).

28. *p. 8 l. 32ff: The authors should indicate which Figure or Table they are referring to.*

The presentation of these results are now within section 3.1, with appropriate reference to data plotted in Fig. 3.

29. *p.9 l. 1f: What is an estimate of this uncertainty then? Can the authors provide a quantification? Otherwise, it is better not to include such statements.*

We removed this sentence.

30. *p.10 l.3 – 18: I do not see the immediate relevance of this discussion. This could be mostly removed to streamline the manuscript and focus on more crucial results, which are given in the next paragraph. I would prefer to see the authors try to include an actual quantitative uncertainty assessment of their measurement method, including the effect of grain shape on their modeling results instead of this ancillary discussion.*

While these results and discussion are not directly relevant to the NERD snow SSA calibration at 1.30 μm, we include them in the manuscript because they are interesting, surprising results. In response to this comment, we trimmed this discussion.

As pointed out, different grain shapes have an effect on the Monte Carlo calculated snow albedo and BRFs. The spread in these calculations is plotted in Figs. 3 and 4. We speculate that these variations across shape habits are directly related to the variations in the particles' asymmetry parameters.

[Figure]

Droxtals have the lowest asymmetry parameters while hexagonal plates have the highest. It is difficult to determine why the simulated BRFs are larger at 30 degrees (zenith) than at 60 degrees. We speculate that the combination of backscatter and more scattering at 30 degrees than at 60 degrees, according to the phase functions, are partially responsible for the different BRFs at 30 versus 60 degrees. Confirming these speculations would require more Monte Carlo testing and further investigation.

While we are unable to reach a conclusion regarding these concerns, we added reference to Kaempfer et al. (2007) who also show larger reflectance factors at 30 degrees than at 60 degrees (viewing) (for $\lambda = 900$ nm).

31. *p. 10 l. 22: I do not see a compelling reason in the presented data why this relationship has to be an exponential function. Could it be something else (linear relationship), and would this significantly change any of the results? If so, this effect should be included in the SSA measurement uncertainty. Also, what does a constant SSA measurement uncertainty across the entire SSA range mean for using the NERD to measure different snow types? High relative SSA measurement uncertainty for snow characterized by low SSA (what snow types are those?) and low relative SSA measurement uncertainty for snow characterized by high SSA (what snow types are those?). This should be added to the text to illustrate the uncertainty in the derived BRF-to-SSA relationship beyond the mere presentation of the values for the root mean square differences.*

These are good discussion points that would improve the presentation of the NERD as an accurate snow SSA instrument. Unfortunately, we do not have enough measurement data yet to fully address these comments. As a result, we changed the focus of the revised manuscript to focus on our key scientific results pertaining to snow metamorphism with and without added LAIs.

32. *p. 11 l. 9: Delete 'accurate'. There is no quantitative validation of the SSA measurements in the manuscript. 'quick, reliable, and repeatable' convey the full picture. Even 'repeatable' could be removed because it is implied by 'reliable'.*

    We removed "accurate" and "reliable" from this sentence.

33. *p. 11 l. 16,17,18: The authors should replace 'will' with 'can' or 'could' or 'may', or preface each of these statements by 'We believe that ....' or 'We intend to use the NERD for ...' or 'The analysis indicates the potential for ....' or similar. Selling such statements as foregone conclusions seems like a far stretch given the limited analysis in the manuscript and the inherent uncertainty of future developments (see also point 1 above).*

    We revised this section to provide conclusive statements in the context of the NERDs limitations. We also suggest further validation to better justify the NERD as a tool to accurately monitor snow SSA.

34. *Caption Figure 3: Snow is not a granular material. Better: ... 'as the snow microstructure gets coarser' or 'is characterized by more rounded shapes'.*

    We removed this figure from the revised manuscript.

We are also grateful for Anonymous Reviewer #2's review, as it has helped develop a better presentation of our study. Like Anonymous Reviewer #3, Reviewer #2 also pointed out an inappropriate use of language and recommended a major revision. We agree with this recommendation and revised, reorganized, and rewrote much of the manuscript accordingly.

In the revised manuscript, the new main focus of how LAIs affect snow metamorphism relieves the need for a lengthy background discussion on the state of the art snow SSA measurement methods. Instead, we present background information in the introduction pertinent to understanding how LAI can possibly affect snow albedo feedbacks. Additionally, we removed technical details from the methods section unrelated to the results and discussion. A condensed presentation of these details is now contained in the appendix. Finally, the reorganized the results and discussion section into two main subsections. First, we present results from our NERD BRF to SSA calibration study, including those from Monte Carlo modeling. Second, we introduce new results from LAI in snow experiments using the NERD to observe snow metamorphism. Because we removed language that implies that the NERD is validated for precise snow SSA retrieval, we emphasize approximate SSA results are enough to observe the significant difference in snow surface behavior in experimental snow with added LAI versus natural snow. The revised discussion therefore focuses on the NERD and our experimental results specific to this study, relieving the need for a lengthy discussion on the advantages and disadvantages compared to state of the art snow SSA measurement methods.

Following are our responses to Reviewer #2's comments, which we italicize for reference:

1. *p.1, L17-18: "Positive snow internal albedo feedback occurs due to the strong dependence of snow infrared reflectance on snow specific surface area (SSA)." This sentence is too compact. Please explain this internal albedo feedback more explicitly.*

   We rewrote the introduction section. As a result, the mention of snow internal albedo feedback is delayed until later in the introduction, where we describe this more specifically in the

context of our revised manuscript.

2. *P1., L18-20: "The Snow, Ice, and ... in Fig. 1". Fig 1 is not sufficiently justified here. It should be moved later in the paper, when describing the reason for the selection of the wavelengths 1300 and 1550 nm for the detection of SSA.*

Thank you for this suggestion. We moved mention of these basic SNICAR modeling results to the methods section, where we describe the motivation for selecting 1300 and 1550 nm.

3. *p.2, L9: "...are equivalent for convex bodies (see Appendix A)." There is no need to write an appendix to make a geometrical demonstration that was already derived more than 150 years ago. Instead, please refer to some book of convex geometry, or better to the original demonstration by Cauchy (as done in Pirazzini et al.: "Measurements and modeling of snow particle size and shortwave infrared albedo over a melting Antarctic ice sheet", The Cryosphere, 9, 2357-2381, https://doi.org/10.5194/tc-9-2357-2015, 2015).*

We removed this appendix. The lone appendix now contains specific details regarding the NERD that we removed from the methods section in response to below comments.

4. *p.2, L16-17: "observe seasonal scale snow albedo decline in springtime Colorado". Could you please improve the expression, for instance as "observe snow albedo decline during the spring season in Colorado"?*

We removed the paragraph containing this sentence.

5. *p.2, L17: "In contrast, however, they find that snow albedo is primarily related to dust concentration." This sentence is incorrect. First of all, the snow albedo is mostly determined by the optical properties of the snow, and not by dust concentration. You may want to say that it is affected by dust concentration, but you cannot claim that it is the main albedo driver. Secondly, why you wrote "In contrast"? In the paper by Skiles and Painter (2017) the springtime albedo decline was accelerated by the dust load, which concentrated at the surface during the progress of the melting further decreasing the albedo. Hence, the increase in dust concentration at the surface affected the observed albedo decline, and was not in contrast with it.*

Thank you for pointing out the confusing style of this sentence. Because this entire paragraph was worded poorly, we removed it from the revised manuscript. In the revised manuscript, general background information regarding how LAIs directly affect snow albedo is rewritten and presented in the first paragraph.

6. *p.2, L19: "where the albedo reduction..." Instead of "where" I suppose you meant something like "who showed that...", right?*

We removed this sentence and relocated the relevant citation to paragraph one (see also above response).

7. *p.2, L21: "snow internal albedo feedback" shouldn't be "internal snow albedo feedback"? As pointed out in my comment above, it is not at all clear what you mean for "internal" snow albedo feedback. Please explain.*

We removed the mention of snow "internal" albedo feedback in the introduction. In the revised manuscript, we relocate this topic to the discussion section where we further describe this snow metamorphism based feedback, where the decrease of snow SSA enhances absorbed infrared radiation which contributes positively to additional snow melt.

8. *p.2, L23-24: "Surface warming can also reduce snow grain growth rates, however, if growth processes from vapor diffusion and strong temperature gradients are affected negatively (Flanner and Zender, 2006)." The meaning of this sentence is very obscure. Could you explain more clearly what you mean, without requiring from the reader to study Flenner and Zender in order to understand what you mean?*

We thoroughly revised the introduction. To this end, we clarified temperature gradient metamorphism in the context of this study.

9. *p.2, L25-31: "Recent studies ..." This section seems to be out of context: it is not linked to the purpose of the paper. Please remove it, or explicitly explain the connection with the content of the paper.*

We reworded this section in the context of the revised manuscript's main purpose. We agree that this section was originally out of context, but in the revised manuscript, it is directly relevant to the main results and discussion.

10. *p.3, L16: "The NERD is designed to measure 1.30 and 1.55 µm BRFs". Please explain here why these wavelengths were selected, and highlight here the analogy with DUFISSS in the wavelength selection.*

We changed the first two paragraphs of section 2.2 to one, explain why these wavelengths were selected, and two to compare the method to the DUFISSS. Accordingly, Fig. 1 is now referred to here to demonstrate the utility of 1.30 and 1.55 µm reflectance measurements in determining snow (with LAIs) grain size. Paragraph two starts by describing the analogy with DUFISSS before stating the NERD technical description.

11. *p.3, L25: "The NERD is similar to that of ... in that it uses ...." Please reformulate the sentence improving the linguistic expression and moving it above (see previous comment).*

We revised the sentence. It now reads "The design principle is similar to the...." "and The NERD also uses.... (sec. 2.1, par. 2)"

12. *p.3, L27-30: "LEDs are toggled ... (20% duty cycle)" A lot of not needed technical details. Please remove.*

We removed these details.

13. *p.3, L31-32: "Here, rather, we direct photodiodes toward the illuminated surface in a black dome to measure BRFs" The linguistic expression is particularly poor in this sentence. Instead of "we" use a passive expression.*

We removed this sentence. Furthermore, the methods section in the revised manuscript is more consistent in the use of passive voice.

14. *p.4, Sect 2.1 and 2.2. Please remove all the technical details that do not provide any added value to the interpretation of the measurements. E.g. "Waiting 0.75 seconds after toggling the LED allows for enough time for the photodiode current to stabilize. After these currents*

*stabilize, 100 voltage samples (ranging from 0.1 to 1.0 Volts) are then rapidly collected using the Ruggeduino-ET's ADCs. The average voltage obtained during active illumination is differenced from the average dark current voltage to derive reflectance 10 factors.", "The reflectance of the targets are measured with high precision across a broad spectrum. At 1.30 (1.55) µm, the white and gray targets have calibrated reflectances of 0.95073 (0.94426) and 0.42170 (0.41343), respectively, as reported by the manufacturer.", "Small samples of snow are collected in roughly 10 cm tall cylindrical plastic sample holders and placed into the machine. An X-ray source is emitted at 40-45 kV and 177-200 micro-Amps. X-ray transmittance is measured as the machine rotates the sample. Setting the exposure time to 340 ms at a pixel resolution of 14.9 µm with rotation steps at 0.3-0.4 degrees allow for fast scan times of roughly 15 minutes. These short scan times are necessary to complete the scan without too much absorbed radiation melting the snow."*

We removed these technical details. We also repurposed the appendix to contain details only relevant to the operation of the instrument. In the revised manuscript, section 2.1 contains details only relevant to the results and discussion and not to preliminary instrument results obtained in testing.

15. *p.4, L19: "Using both…" What do you mean for "both"?*

We meant "two," but reworded this sentence to mitigate confusion.

16. *p.5, Sect 2.3. This section needs to be rewritten in a much more compact and consistent way. Expressions such as "oldest class" are meaningless. You should really apply the snow descriptors listed in The International Classification for Seasonal Snow on the Ground (Fierz, 2009). Instead of repeating 6 times that the measurements were performed in Hanover and samples were transported to CRREL for X-ray microTomography, focus on the characteristics of the different samples. Eliminate subparagraphs and unnecessary details such as "…distinguishable only by the container they were stored in…", or the sentences in lines 24-27 (until "…nearby lab for X-CT analysis"), and the not relevant sentence "All samples with added LAI included in the NERD SSA calibration dataset were first screened to remove samples with heavy LAI loads that caused direct snow darkening at 1.30 5 and 1.55 µm."*

We followed these suggestions and revised the methods section accordingly. These descriptions are now condensed and summarized in Table 2. We classified snow samples according to Fierz et al. (2009). Furthermore, we applied their convention to data plotted in Figs. 4 and 5 so that colors, key codes and symbols conform (as closely as possible) to their snow classification descriptors.

17. *p.6, Sect. 2.4. You need to provide some introduction explaining the purpose of the model simulations*

The purpose of the Monte Carlo simulations is to study light emission by the NERD and the resulting scattering within idealized snow packs. This is now stated at the beginning of section 2.3.

18. *p.6, L30: replace "multiple" with "BRFs"*

We removed this sentence.

19. *p.7, L24 and following: this sections need to be moved after the thorough presentation of the results.*

We rewrote the entire results and discussion section. To this end, we moved all NERD Lambertian test results and analysis to the end of the NERD specific subsection in presented within section 2.

20. *p.8, L12-13: "Both instruments make use of Lambertian reflectance standards for calibration and testing." This is a repetition: you have already explained in the lines above. Please remove it. Instead of discussing the opportunity of using reflectance standards to calibrate SSA detector based on active optical sensors, you could focus the discussion on comparing the different working principles, and strengths/weaknesses of the devices.*

We removed this sentence and instead focus discussion on snow SSA calibration.

21. *p.8, L13-16: "Although each instrument…" This sentence is a rather obvious statement that does not add anything to the paper. Please remove.*

We removed this sentence.

22. *p.8, L23: "Although photodiode responsivity varies with temperature, frequent calibration minimizes these errors" This is a critical point that deserves further explanation. Is calibration required on the field before/after each measurement (as done for instance when using the IceCube device)? If this is the case, please explain, and describe the needed measurement procedure, including calibration.*

Yes, (frequent) calibration is required (preferred) in the field to correct for temperature effects that change photodiode and LED performance. These are more technical points that we removed in the revised manuscript to focus on more relevant experimental procedures. We could, however, provide further discussion in an appendix.

23. *p.8, L34 – p.9, L1: "Monte Carlo simulations predict lower BRF values at 60 degrees than at 30 degrees". This sentence refers to radiances at 1.30µm: looking at Fig. 4 I see the opposite, i.e. that for most grain shapes, when SSA is larger than 40 $m^2kg^{-1}$ BRF is larger at 60 degrees than at 30 degrees.*

Monte Carlo BRFs (line segments) are higher at 30 degrees viewing than at 60 degrees viewing. Oppositely, NERD BRFs are higher at 60 degrees viewing than at 30 degrees viewing. These are important results that we elucidate in the revised results and discussion.

24. *p.9, L17: "Hemispherical reflectance measurements theoretically reduce measurement variations associated with grain shapes". Why? Please explain. Comparing Fig. 4 (top) and Fig. 5 (top) where, respectively, BRFs and directional-hemispherical albedo are illustrated, I would say that both hemispherical and directional measurements show a very similar dependence on grain shape. In my opinion, Fig. 4 would deserve a much deeper analysis. For instance, why the BRFs measured with NERD are so much higher than the model results in 1.30 µm at 60 degrees? And why the modeled BRFs at 1.30 µm are lower at 60deg than at 30 deg? Etc…*

We clarified comparisons of Monte Carlo modeling to the NERD measurements and removed

the confusing sentence quoted above.

25. *p.9, L31-32: "These large variations in reflectance across grain shape are the largest source of uncertainty in snow SSA measurements using infrared reflectance." I disagree. Even larger uncertainties can be associated to the instrument set up in certain snow conditions. You have not discusses the effect of natural light entering into the dome and detected by the photodetectors. Probably, you will have this unwanted light source every time the target snow surface is not perfectly smooth, unless you insert the edges of the dome for several millimeters inside the snow surface. With other optical-based devices to derive SSA (such as DUFISSS and IceCube), a large source of uncertainty may derive from the snow sampling procedure (especially in case of surface hoar or very soft new snow). In my opinion, even your Fig. 4 shows that the large scatter in the optically derived SSA is not only attributable to a grain shape effect. The instrumental and set up error sources deserve much more discussion.*

We removed this sentence. Light entering the dome is subtracted out from background light measurements during each measurement cycle (see also comment # 27 from referee #3).

26. *p.10, L16-17: "These calculations confirm this hypothesis, as 1.55 µm narrow band albedo with a full width at half maximums of 0.26 µm (doubled from 0.13 µm) closely agree with NERD BRF measurements." Please show these results in a Figure.*

We now show these results in Fig. 3 (right).

27. *Figure 2: A much clearer photo of the sensor is needed, which would show only the essential components. The text in the figure should be less technical, or the technical terminology should be explained (what is the meaning of "LCD"? Is the whole sentence "LCD provides … data collection" needed? If yes, you should better explain its content, possibly in the main text and not in the figure. Is the diagram of the Transimpedance amplifier circuit needed? Instead of providing so many technical details, you should explain what the achieved performance is and why it is needed. Also the meaning and scope of the sentence "Using feedback resistances as low as …" in the figure caption is totally obscure. What is the scientific message behind it?*

We removed the circuit diagram and replaced it with a clear photo of the underside of the NERD, which includes the mounted LEDs and photodiodes.

28. *Figure 4: Please mark the vertical and horizontal grids, to facilitate the comparison among the plots.*

We added gridlines to this figure.

29. *Figure 5: what is the added value of this figure? The considerations on the effect of grain shape drown on the basis of directional-hemispherical albedo calculation can equally well been drown on the basis of Fig 4 (showing BRF calculations). I would simply remove the figure.*

We agree that this Figure is slightly redundant. The purpose of the revised figure (now Fig. 3) is twofold: one (left), to compare Monte Carlo albedo calculations directly with SNICAR modeling for snow validation; and two (right), to show how widening the half-widths in

these models supports our hypothesize for why measured BRFs at 1.55 μm are higher than expected (see also comment #26 above).

30. *Table 1: in the table caption please explain the meaning of the used symbols and the content of each column and row.*

We reformatted Table 1 and added symbolic descriptions in the caption.

Finally, we appreciate Dr. Dumont's comments and adhered to their recommendation. We respond to their comments below as done previously:

1. *The introduction is in my opinion, a bit too scattered and confusing and some literature references are also missing. More specifically,*

    (a) *it's a bit weird to have references to calculation in an appendix in the introduction. For me, either the calculation already exists in the literature and then it would be nice to add the reference or it's a new result that should be included in the results section*

    We removed this appendix and moved the reference to Vouk (1948) into the introduction.

    (b) *page 2, lines 22-31 , I don't think it's necessary to go into too much details about the SSA evolution in time, a few sentences without any equation should be sufficient. It's not directly related to the objective of the paper and would give more space for discussing the sate of the art of SSA measurements*

    While these details were irrelevant to the original manuscript, because the main purpose of the revised manuscript is to study how LAIs affect snow albedo feedbacks related to snow metamorphism, this background information is now relevant. We rewrote the introduction accordingly.

    (c) *page 3, lines 1-8, this section is really important for the rest of the paper. It seems to me that it would worth more details on the methods (advantages and drawbacks) and accuracy. Several methodologies are missing here such as IR photography (Matzl and Schneebeli, 2006) , SMP (Proksch et al. 2015) , and retrieval from spectral albedo which is also non destructive (Picard et al., 2016 , Dumont et al., 2017). Regarding SSA calculation from X-ray imaging, I think adding some discussion also on the methodology and resolution issue would be nice (e.g. Hagenmuller et al., 2016).*

    Because we restructured the manuscript to focus on how LAIs affect snow metamorphism, we do not go into too much detail about previous snow SSA measurement techniques. In the revised introduction, we expand on the discussion of only methods directly relevant to this study.

    (d) *Since you also present a Monte-Carlo model, maybe a short state of the art of existing theory and models to simulation snow BRF need to be added and why a new Monte-Carlo model is required ? e.g. Malinka, 2014 , Kokhanovsky and Zege, 2004 , Xi et al., 2006 ....*

    This is a good suggestion, however, we are unsure how to introduce this section into the revised manuscript at this time.

2. *Section 2.1. Some details are missing here (but maybe I did not check carefully enough), what is the diameter of the illumination ? How homogeneous is it ? What is the FOV of the photodiode ? In which azimuthal planes are they with respect to the illumination ?*

We estimate the diameters of the illumination to be 1.5 and 3 cm and added these details to paragraph two of section 2.1. We gathered these details from the manufacture specifications documents which also indicate that the emission patterns are nearly isotropic.

In testing, we detected direct light incident on the photodiodes from the LEDs by observing photodiode current responses when titling the dome upward in a dark room. These tests indicate that both the light emission patterns and the photodiode fields of view are not ideally isotropic and limited to a narrow cone in the forward direction. We eliminated this error source by mounting obstructions in the dome. These obstructions block the direct paths' from LEDs to photodiodes which eliminate the relevant photodiode response.

The exact field of view of the photodiodes is difficult to determine. In the most accurate case, the photodiodes would view a greater surface area than that illuminated by the LEDs. This would ensure that the photodiodes are able to collect most light reflected from subsurface scattering. In the least accurate case, the photodiodes would view a very small surface area of the surface. In this case, while accurate BRFs would still be measurable on surfaces with minimal subsurface scattering, volume scattering would lead to errors in the measured reflectance factors.

Our laboratory testing leads us to believe that the photodiode field of view yields a detectable surface area larger than that illuminated by the LEDs. This enables accurate snow BRF measurements. Because the emission patterns of the LEDs is not ideal, however, there are inevitably imperfections in the BRF measurement. These non-ideal effects are a source of BRF measurement uncertainty.

For more specific details regarding the LEDs and photodiodes, please see the technical documents available from Marktech Optoelectronics: MTE1300N MT51550-IR MTPD1346-100.

3. *Page 7, Equation 5. Here I probably misunderstood something, why is the BRF averaged over all azimuths while the measurement is done only in two azimuthal planes ?*

Modeled BRFs are averaged over all azimuths to reduce Monte Carlo noise and because we expect BRFs to be symmetrical in the azimuthal dimension. This is due to the uniform probability density function for which scattering azimuth angles are randomly generated. Please also see our response to Referee #3's comment # 19.

4. *Section 3, I think it would be less confusing for the reader to start with the model evaluation first.*

We moved the Monte Carlo evaluation to the very beginning of section 3.

5. *Section 3.1. The section is long and a bit confused, can it be re-arranged ?*

Yes. We re-arranged and rewrote all of the results and discussion section.

6. *Section 3.3, comparison with SNICAR should in my mind be part of the model evaluation. It's a bit confusing to have it mixed with the calibration.*

We moved the SNICAR comparison to the model evaluation section presented at the beginning of section 3.

7. *In the discussion, I would also add some details on the surface roughness effects and liquid water effect maybe (e.g. Gallet et al., 2014)*

8. *To my mind, both the conclusions and the abstracts should more clearly state the advantages and drawbacks of this new instrument compared to existing ones. The estimated accuracy in the SSA measurements should also be stated in the abstract.*

   We rewrote both the abstract and conclusions. They now include the NERD SSA uncertainty margin ($10 \text{ m}^2\text{kg}^{-1}$) and brief discussion of the utility of the instrument in the context of the main limitations.

9. *Page 3, line 32, "flat black paint", it would be super interesting to know the spectrum, flat in which range ? I think these details are important for the discussions in the end of the paper.*

   We do not know the spectral characteristics of the flat black paint. We did, however, paint an experimental surface and measured very low BRFs at 1.30 and 1.55 μm, confirming that it is highly absorptive at these wavelengths.

10. *Section 2.2. An accuracy assessment of the SSA calculation from the X-ray images would be nice. I think 14,6 microns is quite rough for snow types of snow (e.g. e and a in Fig. 3).*

    Unfortunately, we do not present these uncertainties here. We did, however, add a reference to Lieb-Lappen et al. (2017) who provide a thorough analysis of the micro-CT procedure. Multiple samples for a given snow type (needles) provide an estimate of this uncertainty. These are indicated by the horizontal error bars in Fig. 4. For needles, this margin of uncertainty appears to be $+/- 10 \text{ m}^2\text{kg}^{-1}$.

11. *Section 2.3 Maybe a table would be clearer than a text description.*

    We summarized this section in Table 2 of the revised manuscript and included a physical classification in response to Referee #2's comments.

12. *Page 6 line 19 scatter →scattering*

    We changed "scatter" to "scattering."

13. *Page 6 line 20. After how many scatter do you stop following the photon ?*

    We do not terminate photons until they are absorbed or exit the snow medium. This is computationally possible at these relatively long NIR wavelengths, but such a scattering events cut off is necessary for simulating shorter wavelengths less than roughly 700 nm.

14. *Page 10 - last section, Picard et al., 2016 and Dumont et al., 2017 provide a detail assessment of the SSA retrieval uncertainties.*

15. *Page 11 – lines 3-7, this should be also indicated in the introduction.*

    We indicate this in the revised introduction.

---

## Referee Report (RR1)

Review of revised manuscript:
Monitoring of snow surface near-infrared bidirectional reflectance factors with added light
absorbing impurities
A. Schneider, M. Flanner, R. De Roo, and A. Adolph

General comments:

I think the authors present a streamlined and much-improved revised version of their manuscript.
They also addressed in great detail the comments I had about the initial manuscript, which I
appreciate. I only have a few minor comments regarding the revised manuscript (see specific
comments below).
While the main focus of the manuscript has shifted (and the manuscript benefits from this), I would
still encourage the authors to tackle a more detailed assessment of the snow SSA measurement
uncertainties for their NERD in the future (as they seem to allude to in the final sentence of their
conclusions). Unfortunately, such uncertainty analyses are still not always provided when a(ny)
novel measurement technique is introduced, yet they can be highly valuable for the application of
a(ny) novel measurement technique, especially when trying to interpret initially puzzling
measurement results from both a qualitative and a quantitative perspective or for an inter-
comparison of different measurement techniques or when comparing in situ measurements and
remote sensing retrievals. For snow SSA measurements with the NERD, one crucial component that
should be included in more detail in a possible future study is how the natural variability of snow at
and near the surface and especially within the NERD measurement volume may affect derived snow
SSA values.

Specific comments:

page 1 line 23: I do not fully understand the different expressions for sphere effective radius $r\_eff$
and Re; is one definition based on the ice surface area and the other one on the projected area?
Maybe the authors could either briefly clarify the difference or only introduce one of the two
effective radii here.

p.2 l.7, 9 and 12: I would suggest to remove the word 'accurate', because the usage of the qualifier
'accurate' should also include information on how accurate the measurement method is (i.e.,
accurate ... with an uncertainty of XYZ % or with an accuracy of better than xyz m2/kg, for
example). If such information about the measurement uncertainty cannot be obtained or
summarized easily, I would just leave out this qualifier.

p.4 l.2ff: To illustrate this point, the authors could provide the first figure that they included in their
author response in a Supplement to the article or in a second Appendix section, or they could
possibly cite a previous study that shows this shallow penetration depth of long-NIR-wavelength
radiation in snow.

p.5 l.1: Is '1 nm' correct? This should probably be 1 $\mu$m (or 1000 nm).

p.6 l.24: What is 'just a pinch' of BC? According to the caption of Figure 7, this seems to be < 1 g. I
would suggest to add this value here as well: …, just a pinch (< 1 g) of BC and 30 g of sand were
deposited …

p.10 l.8: Again, without further specifying 'accurate', e.g., a specific accuracy that the NERD aims

to achieve, I would remove 'accurate' and rewrite the sentence, e.g.: … are needed to fully characterize snow SSA measurements by the NERD (technique). Further investigation …

p.10 l.9: Similarly as above for 'accurate', I would suggest to remove the qualifier 'precise'. Alluding to 'quantitative uncertainties' already implies that the accuracy and precision of snow SSA retrievals will be the subject of the follow-on study.
By the way, I believe that such a study will be very valuable for the future application of the NERD and the interpretation of the measurement results.

Caption of Figure 3: Remove comma before droxtals.

Caption of Figure 6 + 7, and possibly in corresponding text of the article: I would suggest to replace 'standard errors' with 'standard deviations'.

Caption of Figure 7: Are the units of $gm^{-1}$ correct, as in $< 1 \ g \ m^{-1}$ and $30 \ g \ m^{-1}$? Maybe I do not fully understand, but units of $g \ m^{-2}$ would make more sense to me.

---

## Referee Report (RR2)

**Review of "Monitoring of snow surface near-infrared bidirectional reflectance factors with added light absorbing impurities" by Schneider et al.**

I highly appreciate the amount of work that was clearly provided by the authors to address the reviewers comments. The new structure of the paper is way more clear and easy to read and understand than the first version. However, I feel that some points still need to be addressed before it can be published. These points are listed below.

**Specific comments**

Title, abstract and everywhere in the text : LAI can be a misleading acronym, the use of LAP (light absorbing particle) is maybe to be preferred

P1, Line 20 – Picard et al., 2009 did not use only spherical ice particles

P1, Lines 14-15 – LAP can also be living particles, maybe the recent review from Skiles et al., NCC on LAP in snow can be added as reference in the introduction

*Skiles, S. M., Flanner, M., Cook, J. M., Dumont, M., & Painter, T. H. (2018). Radiative forcing by light-absorbing particles in snow. Nature Climate Change*

P2, Line 15 – References to SSA profilers such as ASSSAP, POSSUM or SMP are missing

*Arnaud, L., Picard, G., Champollion, N., Domine, F., Gallet, J.C., Lefebvre, E., Fily, M. and Barnola, J.M., 2011. Measurement of vertical profiles of snow specific surface area with a 1 cm resolution using infrared reflectance: instrument description and validation. Journal of Glaciology, 57(201), pp.17-29.*

*Proksch, M., Löwe, H. and Schneebeli, M., 2015. Density, specific surface area, and correlation length of snow measured by high-resolution penetrometry. Journal of Geophysical Research: Earth Surface, 120(2), pp.346-362.*

P2, lines 16-18 – "in isothermal snow, highly faceted snow grains" this sentence seems a bit weird to me. Isothermal metamorphism and coarsening also happens for non faceted crystals.

P2, lines 25 and below – The beginning of the paragraph is a bit difficult to follow. I agree with the general idea I don't see any clear link with the objective of the paper and I would remove it. I would also reverse the order of the two objectives in accordance with the paper structure.

*General remarks on the introduction:*  I am not all questioning the utility of the instrument and measurements but from the sole information provided by the authors, it is a bit difficult to understand why a new instrument is needed and what are the specifications. Regarding the objective one, I would also recommend that this quantification of snow albedo feedback impact on metamorphism be justified in light of previous studies and measurements.

Maybe start by section 2.3 (modelling) and then 2.2 and 2.4 (two "measurements" sections)

Section 2.2.2 lines 24-26 – Is it possible to provide the absolute changes calculated in SSA, also maybe give explicitly value of tau and n.

Section 2.4.2 what is the approximated mass of dust that was spread on the snow surface? 30g m-2 ? How does it compare to values from Skiles and Painter, 2017?

P7, lines 6-9 – What's the point of the last sentence? It needs to be removed or detailed a bit more.

P7 lines 10-15 – The information discussed here seems quite redundant with section 3.2, is it possible in sake of clarity to remove redundancies?

P7 lines 31 – This is also in line with more theoretical studies such as Kokhanovsky and Zege, 2004 and Malinka, 2014.

*Kokhanovsky, A.A. and Zege, E.P., 2004. Scattering optics of snow. Applied Optics, 43(7), pp.1589-1602.*
*Malinka, A.V., 2014. Light scattering in porous materials: Geometrical optics and stereological approach. Journal of Quantitative Spectroscopy and Radiative Transfer, 141, pp.14-23.*

P9, line 16 "realistic", maybe a bit more details/references is required.

P9, lines 18-19 – "initiated melting" depends on the weather conditions (not only clear/cloudy), and the snow albedo feedbacks is also present before melting.

P3 line 13-14, the authors stated that the measurement is not sensitive to small BC concentrations, but is it sensitive to the large amount of BC or dust used in the experiments ?

**Minor comments**

P1, Line 1 – snow albedo -> broadband snow albedo

P1, Lines 11-12 – the last sentence should maybe be move after measurements (line 8)

P1, Line 9 -10 – "These findings …" as stated in the main text, the results here is not a new finding so maybe rephrase

P1, Line 22 – "its effective radius" -> "its effective radius, Re"

P2, lines 23-24 – I would remove this last sentence,

P3, line 20 – "Flat paint" the details provided in the response to reviewer are maybe useful in the text of the paper too.

P4, line 3 – "at most a couple of centimeters" a few references would be useful for the reader here.

P5, line 1 - "were conducted" -> "were conducted only" ??

P5, line 5 - "at random" -> maybe one word is missing

P5, line 13 - typo for snowpacks

P5, lines 15 -20 – maybe explain why a different choice is conducted for spheres and for the other shapes.

P6 – first paragraph. I am a bit confused by all the different numbers of photons. In the end, 1,000,000 photons was chosen for the simulations ? maybe just rephrase this section.

P6, line 12 – "were sifted" -> which diameter ?

P6, line 22 – I don't think that diffuse radiation is isotropic for cloudy conditions. Maybe rephrase "nearly isotropic".

Figure 5 is quite difficult to read, maybe the model results can be shown in black without markers to ease the comparison with the NERD measurements?

P8, lines 19-22 , "As expected", "typically" : can you provide any reference for that ?

P8, line 33 – "little to no effect" -> during the time of the experiment? "only" 16 hours

Figure 6  and Figure 7 : it is quite difficult to guess what are the limits of the errors bar, can it be  modified ?
Legend of figure 7, the labelling is different for the upper and lower panels, maybe homogenize.

---

## Author Response (AR2)

May 18, 2019

Dear Dr. Chambon,

We are grateful that the referees agreed to review our revised manuscript (tc-2018-198), titled *Monitoring of Snow Surface Near-Infrared Bidirectional Reflectance Factors with Added Light Absorbing [Particles]*, once again. We appreciate their new comments and revised the manuscript accordingly. We believe that their suggestions continue to improve the structure and readability of the paper.

In the following attachment, we respond to the referees' comments, as before. We begin with our response to Dr. Dumont and then address Anonymous Referee #3's comments. Following our point-by-point response is the marked-up manuscript showing our changes, which appear extensive due in large part to the rearranging of paragraphs in the introduction and in section two. This rearrangement is necessary to fully address Dr. Dumont's remarks regarding the unclear motivation for the study as well as their suggestions for improving the consistency within sections. Finally, we carefully checked the manuscript for typos and misnomers, which led to a few changes including added references, updated terminology, and refined equations.

We appreciate your complimentary report and hope that you find our latest revision more polished and ready for publication in *The Cryosphere*.

Sincerely,

Adam Schneider
amschne@umich.edu

We appreciate Dr. Dumont's second review. In addressing their comments, revising continues to improve the overall structure of the paper. Here, Dr. Dumont points out some remaining concerns, the most critical of which addresses the unclear motivation for the objectives stated in the introduction. In the revised manuscript, we rearranged parts of the introduction to narrow in on the foci of the study. This includes a revised paragraph that better presents our motivation, as further described below (see comment 7).

1. *Title, abstract and everywhere in the text : LAI can be a misleading acronym, the use of LAP (light absorbing particle) is maybe to be preferred*
   We changed all instances of light absorbing impurities (LAIs) to light absorbing particles (LAPs), including in the title.

2. *P1, Line 20 – Picard et al., 2009 did not use only spherical ice particles*
   We removed the reference to Picard et al., 2009 from this part of the Introduction, but still cite their similar results in the Results and Discussion.

3. *P1, Lines 14-15 – LAP can also be living particles, maybe the recent review from Skiles et al., NCC on LAP in snow can be added as reference in the introduction*
   *Skiles, S. M., Flanner, M., Cook, J. M., Dumont, M., & Painter, T. H. (2018). Radiative forcing by light-absorbing particles in snow. Nature Climate Change*
   In reference to biological LAPs, we added "microbes" to the list of LAPs in the introduction and also included a reference to Skiles et al., 2018.

4. *P2, Line 15 – References to SSA profilers such as ASSSAP, POSSUM or SMP are missing*
   *Arnaud, L., Picard, G., Champollion, N., Domine, F., Gallet, J.C., Lefebvre, E., Fily, M. and Barnola, J.M., 2011. Measurement of vertical profiles of snow specific surface area with a 1cm resolution using infrared reflectance: instrument description and validation. Journal of Glaciology, 57(201), pp.17-29.*
   *Proksch, M., Löwe, H. and Schneebeli, M., 2015. Density, specific surface area, and correlation length of snow measured by high-resolution penetrometry. Journal of Geophysical Research: Earth Surface, 120(2), pp.346-362.*
   Because the POSSUM is very similar to the NERD, we added a sentence that briefly describes its SSA retrieval method and also cites Arnaud et al., 2011. We also included the POSSUM in the context of our motivation and now reference the authors in the description of the NERD in section 2.2. We are unsure, however, where to include Proksch et al., 2015.

5. *P2, lines 16-18 – "in isothermal snow, highly faceted snow grains" this sentence seems a bit weird to me. Isothermal metamorphism and coarsening also happens for non faceted crystals.*
   We rephrased this sentence, removing the "...faceted snow..." descriptor. We also moved this revised paragraph before the introduction of SSA measurement methods to provide readers context for how snow SSA is used to evaluate snow metamorphism.

6. *P2, lines 25 and below – The beginning of the paragraph is a bit difficult to follow. I agree with the general idea I don't see any clear link with the objective of the paper and I would*

*remove it. I would also reverse the order of the two objectives in accordance with the paper structure.*

We rewrote the beginning of the paragraph to clarify the motivation for our study. As suggested, we also reversed the order of the objectives to align with the rest of the paper.

7. *General remarks on the introduction: I am not all questioning the utility of the instrument and measurements but from the sole information provided by the authors, it is a bit difficult to understand why a new instrument is needed and what are the specifications. Regarding the objective one, I would also recommend that this quantification of snow albedo feedback impact on metamorphism be justified in light of previous studies and measurements.*

To clarify the purpose of the study, we added a couple sentences regarding the limited access to appropriate snow SSA instruments. In doing so, we attempt to find the optimal balance of motivating the need for the new instrument without implying that snow SSA instruments do not exist. We hope that the revised manuscript sufficiently motivates the desire for an inexpensive instrument that can be used to monitor snow surface SSA with added LAPs, but does not suggest that established snow SSA measurement methods are inadequate.

8. *Maybe start by section 2.3 (modeling) and then 2.2 and 2.4 (two "measurements" sections)*

This is a good suggestion that improves the consistency of the overall structure of the paper. We are particularly thankful for this suggestion. We rearranged section two to begin with the numerical simulation methodology.

9. *Section 2.2.2 lines 24-26 – Is it possible to provide the absolute changes calculated in SSA, also maybe give explicitly value of tau and n.*

Yes. This information is now included in the text at the end of section 2.3.1.

10. *Section 2.4.2 what is the approximated mass of dust that was spread on the snow surface? 30g m-2 ? How does it compare to values from Skiles and Painter, 2017?*

Yes. We deposited 30 grams of filtered sand over 1 square meter of snow. This dust flux was the largest dust deposition event observed by Skiles and Painter, 2017. We clarified this a bit in the text.

11. *P7, lines 6-9 – What's the point of the last sentence? It needs to be removed or detailed a bit more.*

For a fixed particle size, differences in Monte Carlo simulated reflectances across particle shape seem to vary with the particle's asymmetry parameter. This intuitive hypothesis is mostly speculative from Monte Carlo results beyond those presented in the manuscript. As such, we removed the sentence in question.

12. *P7 lines 10-15 – The information discussed here seems quite redundant with section 3.2, is it possible in sake of clarity to remove redundancies?*

This is a helpful comment that motivated us to rearrange parts of sections 3.1 and 3.2. In the revised manuscript, section 3.1 presents results strictly from Monte Carlo simulations. In section 3.2, we present snow BRF and SSA measurements and then compare modeling results with measurements. In revising these subsections, we removed redundancies and

improved the continuity of the presentation of results and discussion.

13. *P7 lines 31 – This is also in line with more theoretical studies such as Kokhanovsky and Zege, 2004 and Malinka, 2014.*
    *Kokhanovsky, A.A. and Zege, E.P., 2004. Scattering optics of snow. Applied Optics, 43(7), pp.1589-1602.*
    *Malinka, A.V., 2014. Light scattering in porous materials: Geometrical optics and stereological approach. Journal of Quantitative Spectroscopy and Radiative Transfer, 141, pp.14-23.*
    We added a reference to Malinka, 2014 in the description of the Monte Carlo model with added details to better present our model in the context of similar studies. We also added a reference to Kokhanovsky and Zege, 2004, making note of the agreement between their theoretical framework and our modeling results.

14. *P9, line 16 "realistic", maybe a bit more details/references is required.*
    We changed "realistic" to "extreme" to better summarize our findings.

15. *P9, lines 18-19 – "initiated melting" depends on the weather conditions (not only clear/cloudy), and the snow albedo feedbacks is also present before melting.*
    We removed this speculative sentence.

16. *P3 line 13-14, the authors stated that the measurement is not sensitive to small BC concentrations, but is it sensitive to the large amount of BC or dust used in the experiments ?*
    Large amounts of added BC and dust did have a (small) direct effect on measured BRFs. We added details regarding measurements conducted soon after the initial application of the LAPs to the text, including how much large LAP applications affected BRF measurements.

17. *P1, Line 1 – snow albedo -> broadband snow albedo*
    We added the word "broadband."

18. *P1, Lines 11-12 – the last sentence should maybe be move after measurements (line 8)*
    We moved the last sentence to after the measurement summary.

19. *P1, Line 9 -10 – "These findings ..." as stated in the main text, the results here is not a new finding so maybe rephrase*
    As such, we changed "These findings..." to "These results..."

20. *P1, Line 22 – "its effective radius" -> "its effective radius, Re"*
    We now define "sphere effective radius, $r_e$," verbatim at the beginning of the second paragraph in the introduction.

21. *P2, lines 23-24 – I would remove this last sentence,*
    We removed this sentence.

22. *P3, line 20 – "Flat paint" the details provided in the response to reviewer are maybe useful in the text of the paper too.*

We added a few details regarding how we tested the black paint at the end of the second paragraph in section 2.2.

23. *P4, line 3 – "at most a couple of centimeters" a few references would be useful for the reader here.*
We added references to Smith et al., 2018, Kaempfer et al., 2007, Grenfell et al., 1994, and Brandt and Warren, 1993. They provide further information regarding volume scattering in snowpack.

24. *P5, line 1 - "were conducted" -> "were conducted only" ??*
We conducted contact spectroscopy measurements only for two snow samples (DH_2016 and RG_2015). We rephrased this sentence to clarify this point.

25. *P5, line 5 - "at random" -> maybe one word is missing*
We revised the section (2.1) relevant to this comment and removed the poor phrasing.

26. *P5, line 13 - typo for snowpacks*
We changed all instances of the typo "snow pack" to "snowpack."

27. *P5, lines 15 -20 – maybe explain why a different choice is conducted for spheres and for the other shapes.*
We added the detail that we use the HG phase function for spheres to improve computational efficiency.

28. *P6 – first paragraph. I am a bit confused by all the different numbers of photons. In the end, 1,000,000 photons was chosen for the simulations ? maybe just rephrase this section.*
There are indeed a lot of photons. We revised this paragraph, now at the end of section 2.1, to hopefully mitigate any lingering confusion.

29. *P6, line 12 – "were sifted" -> which diameter ?*
We added details regarding how particles were filtered (roughly 1 mm diameter).

30. *P6, line 22 – I don't think that diffuse radiation is isotropic for cloudy conditions. Maybe rephrase "nearly isotropic".*
We removed the term "isotropic," instead using only "diffuse."

31. *Figure 5 [4?] is quite difficult to read, maybe the model results can be shown in black without markers to ease the comparison with the NERD measurements?*
We changed the colors of the modeling results in figures 3 and 4 to black, for Monte Carlo results, and gray, for SNICAR results. We kept the markers, however, to more easily distinguish between shape habits.

32. *P8, lines 19-22 , "As expected", "typically" : can you provide any reference for that ?*
We cannot provide any references. As such, we removed language suggesting that higher snow SSA derived from contact spectroscopy than from X-CT is an expected result.

33. *P8, line 33 – "little to no effect" -> during the time of the experiment? "only" 16 hours*
We clarified that the results from this experiment pertain only to the 16 hours during which it lasted.

34. *Figure 6 and Figure 7 : it is quite difficult to guess what are the limits of the errors bar, can it be modified ?*
We shifted data in figures 6 and 7 by 15 minutes, for added BC experiments, and by 30 minutes, for added dust experiments. This actually better represents measurement data, as roughly 15 minutes elapsed between data collection between these snow plots.

35. *Legend of figure 7, the labelling is different for the upper and lower panels, maybe homogenize.*
We homogenized the labeling, as suggested.

We are grateful that Anonymous Referee #3 decided to review our revised manuscript, as their first review motivated us to change the main foci of the paper. Without their thoughtful suggestions, the presentation of our key results would have been obscured by an ineffective writing style. We are glad that they believe the revised manuscript is improved and wish to give credit to them for helping us better communicate our results. Below, we respond to their enumerated comments.

1. *I think the authors present a streamlined and much-improved revised version of their manuscript. They also addressed in great detail the comments I had about the initial manuscript, which I appreciate. I only have a few minor comments regarding the revised manuscript (see specific comments below).*
Thank you for reviewing our manuscript(s)! We addressed the below specific comments in a second revision.

2. *While the main focus of the manuscript has shifted (and the manuscript benefits from this), I would still encourage the authors to tackle a more detailed assessment of the snow SSA measurement uncertainties for their NERD in the future (as they seem to allude to in the final sentence of their conclusions). Unfortunately, such uncertainty analyses are still not always provided when a(ny) novel measurement technique is introduced, yet they can be highly valuable for the application of a(ny) novel measurement technique, especially when trying to interpret initially puzzling measurement results from both a qualitative and a quantitative perspective or for an inter- comparison of different measurement techniques or when comparing in situ measurements and remote sensing retrievals. For snow SSA measurements with the NERD, one crucial component that should be included in more detail in a possible future study is how the natural variability of snow at and near the surface and especially within the NERD measurement volume may affect derived snow SSA values.*
While this comment mainly concerns future work, in the conclusions, we added "...investigation of the natural variability of snow near the surface" to the list of topics to include in a follow on study.

3. *page 1 line 23: I do not fully understand the different expressions for sphere effective radius r_eff and Re; is one definition based on the ice surface area and the other one on the projected area? Maybe the authors could either briefly clarify the difference or only introduce one of the two effective radii here.*

Yes. The definitions differ based on the surface area versus projected area. To eliminate potential confusion, as suggested, we now only introduce one effective radius ($r_e$) defined by the surface area.

4. *p.2 l.7, 9 and 12: I would suggest to remove the word 'accurate', because the usage of the qualifier 'accurate' should also include information on how accurate the measurement method is (i.e., accurate ... with an uncertainty of XYZ % or with an accuracy of better than xyz m2/kg, for example). If such information about the measurement uncertainty cannot be obtained or summarized easily, I would just leave out this qualifier.*
Throughout the manuscript, we removed all instances of the qualifier "accurate" when used without corresponding uncertainty quantification.

5. *p.4 l.2ff: To illustrate this point, the authors could provide the first figure that they included in their author response in a Supplement to the article or in a second Appendix section, or they could possibly cite a previous study that shows this shallow penetration depth of long-NIR-wavelength radiation in snow.*
We cited previous studies by Kaempfer et al., 2007, Smith et al., 2018, Grenfell et al., 1993, and Brandt and Warren, 1993, from which the shallow penetration depth at longer NIR wavelengths can be inferred. If the editor considers it necessary, we will include a supplement that further illustrates this point, as suggested.

6. *p.5 l.1: Is '1 nm' correct? This should probably be 1 μm (or 1000 nm).*
Thank you for brining this to our attention. Yes, it should indeed be 1 μm. We corrected this mistake.

7. *p.6 l.24: What is 'just a pinch' of BC? According to the caption of Figure 7, this seems to be < 1 g. I would suggest to add this value here as well: ..., just a pinch (< 1 g) of BC and 30 g of sand were deposited ...*
As suggested, we inserted "(< 1 g)" as a rough quantification of the mass of BC used.

8. *p.10 l.8: Again, without further specifying 'accurate', e.g., a specific accuracy that the NERD aims to achieve, I would remove 'accurate' and rewrite the sentence, e.g.: ... are needed to fully characterize snow SSA measurements by the NERD (technique). Further investigation ...*
Throughout the manuscript, we removed all instances of the qualifier "accurate" when used without corresponding uncertainty quantification (copied from response to comment 4, above).

9. *p.10 l.9: Similarly as above for 'accurate', I would suggest to remove the qualifier 'precise'. Alluding to 'quantitative uncertainties' already implies that the accuracy and precision of snow SSA retrievals will be the subject of the follow-on study.*
*By the way, I believe that such a study will be very valuable for the future application of the NERD and the interpretation of the measurement results.*
We removed this instance of "precise."
Thank you! We are excited about the future development of the NERD and hope to continue with a refined quantitative uncertainty analysis, as suggested (and needed).

10. *Caption of Figure 3: Remove comma before droxtals.*
Removed.

11. *Caption of Figure 6 + 7, and possibly in corresponding text of the article: I would suggest to replace 'standard errors' with 'standard deviations'.*
Because each BRF measurement already averages 100 samples, we believe calculating standard deviations across the NERD BRF measurements are actually better described as "standard errors."

12. *Caption of Figure 7: Are the units of gm-1 correct, as in < 1 g m-1 and 30 g m-1? Maybe I do not fully understand, but units of g m-2 would make more sense to me.*
Thank you for catching this. We corrected these units to $gm^{-2}$.

[revised manuscript text omitted]